# Efficient Discrete Physics-informed Neural Networks for Solving Evolutionary Partial Differential Equations

## Abstract

Physics-informed neural networks (PINNs) have shown promising potential for solving partial differential equations (PDEs) using deep learning. However, PINNs face training difficulties for evolutionary PDEs, particularly for dynamical systems whose solutions exhibit multi-scale or turbulent behavior over time. The reason is that PINNs may violate the temporal causality property since all the temporal features in the PINNs loss are trained simultaneously. This paper proposes to use implicit time differencing schemes to enforce temporal causality, and use transfer learning to sequentially update the PINNs in space as surrogates for PDE solutions in different time frames. The evolving PINNs are better able to capture the varying complexities of the evolutionary equations, while only requiring minor updates between adjacent time frames. Our method is theoretically proven to be convergent if the time step is small and each PINN in different time frames is well-trained. In addition, we provide state-of-the-art (SOTA) numerical results for a variety of benchmarks for which existing PINNs formulations may fail or be inefficient. We demonstrate that the proposed method improves the accuracy of PINNs approximation for evolutionary PDEs and improves efficiency by a factor of 4–40x. All code and data can be found in the supplemental materials.

## 1 Introduction

Evolutionary partial differential equations (PDEs) are representative of the real world, such as the Navier–Stokes equation, Cahn–Hilliard equations, wave equation, Korteweg–De Vries equation, etc., which arise from physics, mechanics, material science, and other computational science and engineering fields Dafermos & Pokorny (2008). Due to the inherent universal approximation capability of neural networks and the exponential growth of data and computational resources, neural network PDE solvers have recently gained popularity Raissi et al. (2017); Han et al. (2018); Khoo et al. (2021); Yu & E (2018); Sirignano & Spiliopoulos (2018); Long et al. (2018). The most representative approach among these neural network PDE solvers is Physics-Informed Neural Networks (PINNs) Raissi et al. (2019). PINNs have been utilized effectively to solve PDE problems such as the Poisson equation, Burgers equation, and Navier-Stokes equation Raissi et al. (2019); Lu et al. (2021a); Mishra & Molinaro (2023). Many variants of PINNs include loss reweighting Wang et al. (2021a; 2022b;a); Krishnapriyan et al. (2021), novel optimization targets Jagtap et al. (2020); Kharazmi et al. (2021), novel architectures Jagtap et al. (2020); Jagtap & Karniadakis (2021); Wang et al. (2021b) and other techniques such as transfer learning and meta-learning Goswami et al. (2020); Liu et al. (2022b), have also been explored to enhance training and test accuracy.

When we apply neural networks to solve evolutionary PDEs, the most ubiquitously used PINN implementation at present is the meshless, continuous-time PINN in Raissi et al. (2019). However, training (i.e., optimization) is still the primary challenge when employing this approach, particularly for dynamical systems whose solutions exhibit strong non-linearity, multi-scale features, and high sensitivity to initial conditions, such as the Kuramoto-Sivashinsky equation and the Navier-Stokes equations in the turbulent regime. Recently Wang et al. Wang et al. (2022a) revealed that continuous-time PINNs can violate the so-called *temporal causality* property, and are therefore prone to converge to incorrect solutions. Temporal causality requires that models should be sufficiently trained at time $t$ before approximating the solution at the later time $t + \Delta t$, while continuous-time PINNs are trained

for all time $t$ simultaneously. To enhance the temporal causality in the training process, they proposed a simple re-formulation of PINNs loss functions as shown in equation 1, i.e., a clever weighting technique that is inversely exponentially proportional to the magnitude of cumulative residual losses from prior times. This casual PINN method has been demonstrated to be effective for some difficult problems. However their method is sensitive to the new causality hyper-parameter $\epsilon$, and the training time is substantially longer than vanilla PINNs.

$$\mathcal{L}(\theta) = \frac{1}{N_t} \sum_{i=1}^{N_t} w_i \mathcal{L}(t_i, \theta), \quad \text{with} \quad w_i = \exp\left(-\epsilon \sum_{k=1}^{i-1} \mathcal{L}(t_k, \theta)\right). \tag{1}$$

In this paper, we introduce a new PINN implementation technique for efficiently and precisely solving evolutionary PDEs. Our technique relies on two key elements: **(a)** using discrete-time PINNs instead of continuous-time PINNs to satisfy the principle of temporal causality, thereby making the training process stable and accurate; and **(b)** utilizing transfer learning to accelerate PINN training in later time frames. The time-differencing schemes such as forward/backward Euler, Crank-Nicolson, and Runge-Kutta enable solutions to be learned from earlier times to later times, therefore satisfying the temporal causality principle. Moreover, the errors from time differencing can be theoretically controlled Ascher (2008), making the training procedure stable and accurate. We accelerate PINN training naturally by initializing the PINN parameters at the next time frame with the trained PINN parameters at the current time frame. In the following sections, we will show that our transfer learning enhanced discrete physics-informed neural networks (TL-DPINN) method is theoretically and numerically stable, accurate, and efficient.

Following is a summary of the contribution of the paper.

- Implicit time differencing with the transfer-learning tuned PINN provides more accurate and robust predictions of evolutionary PDEs' solutions while retaining a low computational cost.
- We prove theoretically the error estimation result of our TL-DPINN method, indicating that TL-DPINN solutions converge as long as the time step is small and each PINN in different time frames is well trained.
- Through extensive numerical results, we demonstrate that our method can attain state-of-the-art (SOTA) performance among various PINN frameworks in a trade-off between accuracy and efficiency.

## 2 RELATED WORKS

**Discrete PINN.** Raissi et al. Raissi et al. (2019) have applied the general form of Runge–Kutta methods with arbitrary $q$ stages to the evolutionary PDEs. However, only an implicit Runge-Kutta scheme with $q = 100$ stages and a single large time step $\Delta t = 0.8$ are computed. Low-order methods cannot retain their predictive accuracy for large time steps. In our research, we demonstrate the capability of discrete PINNs both theoretically and experimentally, indicating that robust low-order implicit Runge-Kutta combined with PINN can obtain high-precision solutions with multiple small-sized time steps. Jagtap and Karniadakis Jagtap & Karniadakis (2021) propose a generalized domain decomposition framework that allows for multiple sub-networks over different subdomains to be stitched together and trained in parallel. However, it is not causal and has the same training issues as conventional PINNs. The implicit Runge-Kutta scheme combined with PINN has been used to solve simple ODE systems Stiasny et al. (2021); Moya & Lin (2023), but not dynamic PDE systems with multi-scale or turbulent behavior over time.

**Temporal decomposition.** Diverse strategies have been studied for enhancing PINN training by splitting the domain into numerous small "time-slab". Wight and Zhao L. Wight & Zhao (2021) propose an adaptive time-sampling strategy to learn solutions from the previous small time domain to the whole time domain. However, collocation points are costly to add, and the computational cost rises. This time marching strategy has been enhanced further in Krishnapriyan et al. (2021); Mattey & Ghosh (2022); McClenny & Braga-Neto (2023). Nevertheless, causality is only enforced on the scale of the time slabs and not inside each time slab, thus the convergence can not be theoretically guaranteed. A unified framework for causal sweeping strategies for PINNs is summarized in Penwarden et al. (2023). Wang et al. Wang et al. (2022a) introduced a simple causal weight in the form of equation 1

to naturally match the principle of temporal causality with high precision. However, this significantly increased computational costs and did not guarantee convergence Penwarden et al. (2023). Our methods can attain the same level of precision, are theoretically convergent, and are 4 to 40 times quicker.

**Transfer learning.** Transfer-learning has been previously combined with various deep-learning models for solving PDEs problems, such as PINN for phase-field modeling of fracture Goswami et al. (2020), DeepONet for PDEs under conditional shift Goswami et al. (2022), DNN-based PDE solvers Chen et al. (2021), PINN for inverse problems Xu et al. (2023), one-shot transfer learning of PINN Desai et al. (2022), and training of CNNs on multi-fidelity data Song & Tartakovsky (2022). Xu et al. Xu et al. (2022) proposed a transfer learning enhanced DeepONet for the long-term prediction of evolution equations. However, their method necessitates a substantial amount of training data from traditional numerical methods. In contrast, our methods are physics-informed and do not require additional training data.

## 3 NUMERICAL METHOD

**Problem set-up**  Here we consider the initial-boundary value problem for a general evolutionary parabolic differential equation. The extension to hyperbolic equations are straightforward.

$$\begin{cases} u_t = \mathcal{N}(u), & x \in \Omega, t \in [0, T], \\ u(0, x) = u_0(x), & x \in \Omega, \\ u(t, x) = g(t, x), & t \in [0, T], x \in \partial\Omega, \end{cases} \tag{2}$$

where $u(t, x)$ denotes the hidden solution, $t$ and $x$ represent temporal and spatial coordinates respectively, $\mathcal{N}(u)$ denotes a differential operator (for example, $\mathcal{N}(u) = u_{xx}$ for the simplest Heat equation), and $\Omega \subset \mathbb{R}^D$ is an open, bounded domain with smooth boundary $\partial\Omega$. This study assumes that the equations are dissipative in the sense that $\int_\Omega u \cdot \mathcal{N}(u)dx \leq 0$ Xu et al. (2022).

Our goal is to learn $u(t, x)$ by neural network approximation. We briefly mention the basic background of PINN in Section 3.1 and then describe our TL-DPINN method in Section 3.2.

### 3.1 PHYSICS-INFORMED NEURAL NETWORKS

In the original study of PINNs Raissi et al. (2019), it approximates $u(t, x)$ to equation 2 using a deep neural network $u_\theta(t, x)$, where $\theta$ represents the neural network's parameters (e.g., weights and biases). Consequently, the objective of a vanilla PINN is to discover the $\theta$ that minimizes the physics-based loss function:

$$\mathcal{L}(\theta) = \lambda_b \mathcal{L}_b(\theta) + \lambda_u \mathcal{L}_u(\theta) + \lambda_r \mathcal{L}_r(\theta), \tag{3}$$

where $\mathcal{L}_b(\theta) = \frac{1}{N_b} \sum_{i=1}^{N_b} \|u_\theta(t_b^i, x_b^i) - g(t_b^i, x_b^i)\|^2$, $\mathcal{L}_u(\theta) = \frac{1}{N_u} \sum_{i=1}^{N_u} \|u_\theta(0, x_t^i) - u_0(x_t^i)\|^2$ and $\mathcal{L}_r(\theta) = \frac{1}{N_r} \sum_{i=1}^{N_r} \|\mathcal{R}(u_\theta(t_r^i, x_r^i)\|^2$. The $t_b^i, x_b^i, x_t^i$ represent the boundary and initial sampling data for $u_\theta(t, x)$, whereas $t_r^i, x_r^i$ represent the data points utilized to calculate the residual term $\mathcal{R}(u) = u_t - \mathcal{N}(u)$. The coefficients $\lambda_b$, $\lambda_u$, and $\lambda_r$ in the loss function are utilized to assign a different learning rate, which can be specified by humans or automatically adjusted during training Wang et al. (2021a; 2022b). We note that the $\mathcal{L}_b$ term can be further omitted if we apply hard constraint in the PINN's design Lu et al. (2021b); Liu et al. (2022a); Sukumar & Srivastava (2022).

As demonstrated in Wang et al. (2022a), the vanilla PINN may violate the principle of temporal causality, as the residual loss at the later time may be minimized even if the predictions at previous times are incorrect. Figure 1 demonstrates the training result for solving the Allen-Chan equation, confirming this phenomenon. For conventional PINN, the residual loss $\mathcal{L}_r$ is quite large near the initial state and decays quickly to a small value when the learned solution is incorrect. Comparatively, our method's residual remains small for all $t \in [0, 1]$ and captures the solution with high precision.

### 3.2 TRANSFER LEARNING ENHANCED DISCRETE PINN

**Discrete PINN**  Since the continuous-time PINN violates temporal causality, we shift to numerical temporal differencing schemes that naturally respect temporal causality. Given a time step $\Delta t$, assume

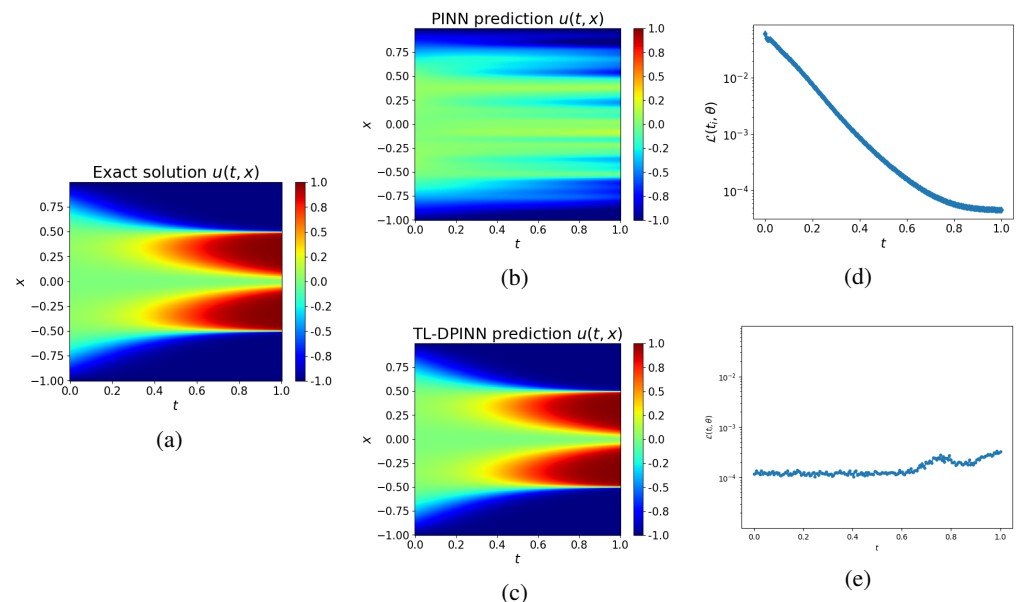

Figure 1: Allen-Cahn equation: (a)reference solution. (b)PINN solution. (c)TL-DPINN solution. (d)PINN's temporal residual loss $\mathcal{L}_r(t_n, \theta)$. (e)TL-DPINN's temporal residual loss $\mathcal{L}_r(t_n, \theta)$.

we have computed $u^n(x)$ to approximate the solution $u(n\Delta t, x)$ to equation 2, then we consider finding $u^{n+1}(x)$ by the Crank-Nicolson time differencing scheme:

$$\frac{u^{n+1}(x) - u^n(x)}{\Delta t} = \mathcal{N}\left[\frac{u^{n+1}(x) + u^n(x)}{2}\right]. \tag{4}$$

Instead of solving equation 2 in the whole space-temporal domain directly, our goal is to solve equation 4 from one step to the next in the space domain: $u_0(x) \mapsto u^1(x) \mapsto \cdots \mapsto u^n(x) \mapsto u^{n+1}(x) \mapsto \cdots$, so that the evolutionary dynamics can be captured over a long time horizon.

Next, we apply PINN to solve equation 4. It is also called discrete PINN in Raissi et al. (2019) when the Crank-Nicolson scheme is replaced by implicit high-order Runge-Kutta schemes. Assuming we have obtained a neural network $u_{\theta^n}(x)$ to approximate $u(n\Delta t, x)$ in equation 2, we compute $u_{\theta^{n+1}}(x)$ by finding another new $\theta^{n+1}$ that minimize the loss functions

$$\mathcal{L}^{n+1}(\theta^{n+1}) = \frac{\lambda_b}{N_b} \sum_{i=1}^{N_b} \left| u_{\theta^{n+1}}(x_b^i) - g(x_b^i) \right|^2$$

$$+ \frac{\lambda_r}{N_r} \sum_{i=1}^{N_r} \left| \frac{u_{\theta^{n+1}}(x_r^i) - u_{\theta^n}(x_r^i)}{\Delta t} - \mathcal{N}\left[\frac{u_{\theta^{n+1}}(x_r^i) + u_{\theta^n}(x_r^i)}{2}\right] \right|^2. \tag{5}$$

These multiple PINNs $u_{\theta^n}(x)$ take $x$ as input and output the solution values at different timestamps.

**Remark 3.1.** *We remark that there exist alternative options for time differencing beyond the second-order Crank-Nicolson scheme. Several implicit Runge-Kutta schemes, including the first-order backward Euler scheme and the fourth-order Gauss-Legendre scheme, have been found to be effective. The second-order Crank-Nicolson scheme is favored due to its optimal trade-off between computational efficiency and numerical accuracy. A comprehensive exposition of these techniques is available in Appendix A.2.*

**Transfer learning** The transfer learning methodology is utilized to expedite the training procedure between two adjacent PINNs. All the PINNs $u_{\theta^n}(x)$ share the same neural network architectures with different parameters $\theta^n$. For a small time step $\Delta t$, there are little difference between the two adjacent PINNs $u_{\theta^n}(x)$ and $u_{\theta^{n+1}}(x)$. So the parameters $\theta^{n+1}$ to be trained are very close to the trained parameters $\theta^n$. The approach involves freezing a significant portion of the well-trained $u_{\theta^n}(x)$

and solely updating the weights in the last hidden layer through the application of a comparable physics-informed loss function equation 5.

To be more precise, we first approximate the initial condition $u_0(x)$ by the neural network $u_{\theta^0}(x)$, then learn $u_{\theta^1}(x), u_{\theta^2}(x), \ldots$ sequentially by transfer learning. The general structure of our TL-DPINN method is illustrated in Algorithm 1.

---

**Algorithm 1:** The training procedure of our TL-DPINN method

**Input** : Target evolutionary PDE equation 2; initial network $u_\theta$; end time $T$
**Output** : The predicted model $u_{\theta^n}(x)$ at each timestamp $t_n$

1  **Set hyper-parameters:** timestamps number $N_t$, number of maximum training iterations $M_0, M_1$, learning rate $\eta$, threshold value $\epsilon$ ;
2  **Step (a):** learn $u_{\theta^0}(x)$ by PINN ;
3  **for** $i = 1, 2, ..., M_0$ **do**
4  $\quad$ Compute the mean square error loss $\mathcal{L}^0(\theta^0)$;
5  $\quad$ Update the parameter $\theta^0$ via gradient descent $\theta_{i+1}^0 = \theta_i^0 - \eta\nabla\mathcal{L}^0(\theta_i^0)$ ;
6  **Step (b):** denote $\theta_*^0 = \theta_{M_0}^0$ and learn $u_{\theta^1}(x), ..., u_{\theta^n}(x), ...$ sequentially by transfer learning ;
7  **for** $n = 0, 1, 2, ..., N_t - 1$ **do**
8  $\quad$ **for** $i = 1, 2, ..., M_1$ **do**
9  $\quad\quad$ Compute loss $\mathcal{L}_i^{n+1}(\theta_i^{n+1})$ by equation 5 ;
10 $\quad\quad$ Update the parameter $\theta^{n+1}$ via gradient descent $\theta_{i+1}^{n+1} = \theta_i^{n+1} - \eta\nabla\mathcal{L}^{n+1}(\theta_i^{n+1})$ ;
11 $\quad\quad$ **if** $|\mathcal{L}^{n+1}(\theta_{i+1}^{n+1}) - \mathcal{L}^{n+1}(\theta_i^{n+1})| < \epsilon$ **then**
12 $\quad\quad\quad$ denote $\theta_*^{n+1} = \theta_i^{n+1}$ and **break** ;

13 **Return** the optimized neural network parameters $\theta_*^1, \theta_*^2, ..., \theta_*^{N_t}$.

---

## 4 THEORETICAL RESULT

In this section, we analyze the TL-DPINN method and give an error estimate result to approximate the evolutionary differential equation 2. We have two reasonable assumptions as follows.

**Assumption 4.1.** *The equation equation 2 is dissipative, i.e. $\int_\Omega u \cdot \mathcal{N}(u)dx \leq 0$ for all $u(t, x)$. Moreover, if $\mathcal{N}$ is nonlinear, then $\int_\Omega(u_1 - u_2) \cdot (\mathcal{N}(u_1) - \mathcal{N}(u_2))dx \leq 0$ for all $u_1(t, x)$ and $u_2(t, x)$.*

**Assumption 4.2.** *The solution $u(t, x)$ to equation 2 and the neural network solution $u_{\theta^n}(x)$ to equation 5 are all smooth and bounded, as well as their high order derivatives.*

The first assumption is to guarantee that the solution is not increasing over time. Consider the $L^2$ norm $\|u(t, \cdot)\|^2 = \int_\Omega u(t, x)^2 dx$, we multiply equation 2 by $u$ and integrate in $x$ to get $\frac{1}{2}\frac{d}{dt}\|u\|^2(t) = \int_\Omega u \cdot \mathcal{N}u dx \leq 0$, so $\|u(t, \cdot)\| \leq \|u_0\|$ for all $t > 0$. For the simplest Heat equation with $\mathcal{N}(u) = u_{xx}$, it is easy to verify that $\int_\Omega u \cdot \mathcal{N}(u)dx = -\int_\Omega |u_x|^2 dx \leq 0$, satisfying Assumption 4.1.

The second assumption can be verified by the standard regularity estimate result of PDEs Evans (2022), and we omit it here for brevity.

Denote the symbol $\tau = \Delta t$ and $t_n = n\tau$, we show that the error can be strictly controlled by the time step $\tau$, the training loss value $\mathcal{L}^n$ and the collocation points number $N_r$.

**Theorem 4.1.** *With the assumptions equation 4.1 and equation 4.2 hold, then the error between the solution $u(t_n, x)$ to equation 2 and the neural network solution $u_{\theta^n}(x)$ to equation 5, i.e., $e^n(x) = u(t_n, x) - u_{\theta^n}(x)$, can be estimated in the $L^2$ norm by*

$$\|e^n\| \leq C\sqrt{1 + t_n}(\tau^2 + \max_{1 \leq i \leq n}\sqrt{\mathcal{L}^i} + N_r^{\frac{1}{4}}), \quad n = 1, ..., N_t, \tag{6}$$

*where $C$ is a bounded constant depend on $u(t_n, x)$ and $u_{\theta^n}(x)$.*

The proof of Theorem 4.1 can be found in Appendix A.3.

Table 1: A comparison of the relative $L^2$ error and training time (seconds) for different PDEs.

| Method | RD Eq. | | AC Eq. | | KS Eq. | | NS Eq. | |
|---|---|---|---|---|---|---|---|---|
| | $L^2$ error | time | $L^2$ error | time | $L^2$ error | time | $L^2$ error | time |
| Original PINN | 4.17e-02 | 1397 | 8.23e-01 | 1412 | 1.00e+00 | - | 1.32e+00 | - |
| Adaptive sampling | 1.65e-02 | 1561 | 8.64e-03 | 1460 | 9.98e-01 | 6901 | 8.45e-01 | 25385 |
| Self-attention | 1.14e-02 | 1450 | 1.05e-01 | 1770 | 8.22e-01 | 5415 | 9.28e-01 | 21296 |
| Time marching | 3.98e-03 | 3215 | 2.01e-02 | 3715 | 8.02e-01 | 5527 | 8.85e-01 | 26200 |
| Causal PINN | 3.99e-05 | 7358 | 1.66e-03 | 9264 | 4.16e-02 | 22029 | 4.73e-02 | 5 days |
| TL-DPINN1 (ours) | **1.82e-05** | 1463 | **5.92e-04** | 2328 | **7.17e-03** | 5050 | **3.44e-02** | 12440 |
| TL-DPINN2 (ours) | 9.34e-05 | **748** | 9.82e-04 | **1100** | 3.55e-02 | 5171 | 3.66e-02 | 56875 |

## 5 COMPUTATIONAL RESULTS

This section compares the accuracy and training efficiency of the TL-DPINN approach to existing PINN methods using various key evolutionary PDEs, including the Reaction-Diffusion (RD) equation, Allen-Cahn (AC) equation, Kuramoto–Sivashinsky (KS) equation, Navier-Stokes (NS) equation. All the code is implemented in JAX Bradbury et al. (2018), a framework that is gaining popularity in scientific computing and deep learning. In all examples, the activation function is $\tanh(\cdot)$ and the optimizer is Adam Kingma & Ba (2014). Appendix A.4.1 discusses the Fourier feature embedding and modified fully-connected neural networks used in Wang et al. (2022a). Appendix A.4.2 details the error metric, neural network hyper-parameters, and training approach.

The Crank-Nicolson time differencing is denoted as TL-DPINN1, while the Gauss-Legendre time differencing is denoted as TL-DPINN2. Our study involves a comparison of these methods with several robust baselines: 1) original PINN Raissi et al. (2019); 2) adaptive sampling L. Wight & Zhao (2021); 3) self-attention McClenny & Braga-Neto (2023); 4) time marching Mattey & Ghosh (2022) and 5) causal PINN Wang et al. (2022a) Table 1 summarizes a comparison of the relative $L^2$ error and running time (seconds) for different equations by different methods. We note that all neural networks are trained on an NVIDIA GeForce RTX 3080 Ti graphics card.

### 5.1 REACTION-DIFFUSION EQUATION

This study begins with the Reaction-Diffusion (RD) equation, which is significant to nonlinear physics, chemistry, and developmental biology. We consider the one-dimensional Reaction-Diffusion equation with the following form:

$$\begin{cases} u_t = d_1 u_{xx} + d_2 u^2, & t \in [0,1], x \in [-1,1], \\ u(0,x) = \sin(2\pi x)(1 + \cos(2\pi x)), \\ u(t,-1) = u(t,1) = 0, \end{cases} \tag{7}$$

where $d_1 = d_2 = 0.01$. The solution changes slowly over time, and Table 1 demonstrates that all methods succeed with small relative $L^2$ norm error in this instance. Our methods enhance accuracy by 2 3 orders of magnitude compared to other PINN frameworks Raissi et al. (2019); L. Wight & Zhao (2021); McClenny & Braga-Neto (2023); Mattey & Ghosh (2022) even with less training time. We see that our method TL-DPINN1 is more accurate than causal PINN Wang et al. (2022a) with much less computational time. We acknowledge that our methods TL-DPINN2 may be slightly less accurate than causal PINN, but the training time is only nearly $1/10$ of their method. In fact, the casual PINN can only achieve a relative $L^2$ error of $1.13e-01$ if we stop early at the training time of our methods ( 748 seconds). Figure 2 shows the predicted solution against the reference solution, and our proposed method achieves a relative $L^2$ error of $1.82e-05$. More computational results of the RD equation are provided in Appendix A.4.3.

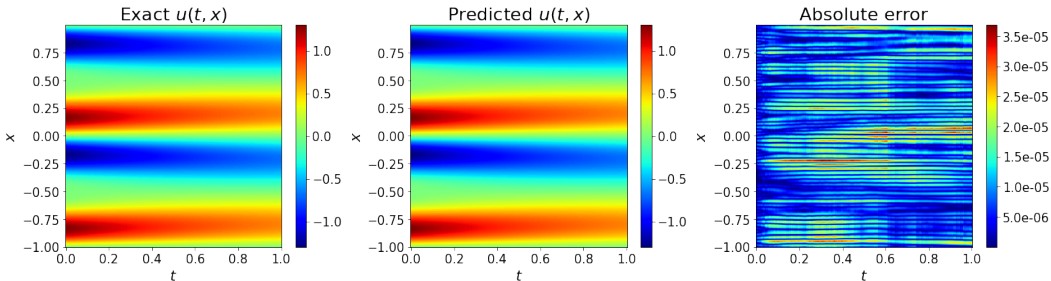

Figure 2: Comparison between the reference and predicted solutions for the Reaction-Diffusion equation, and the $L^2$ error is $1.82e - 05$.

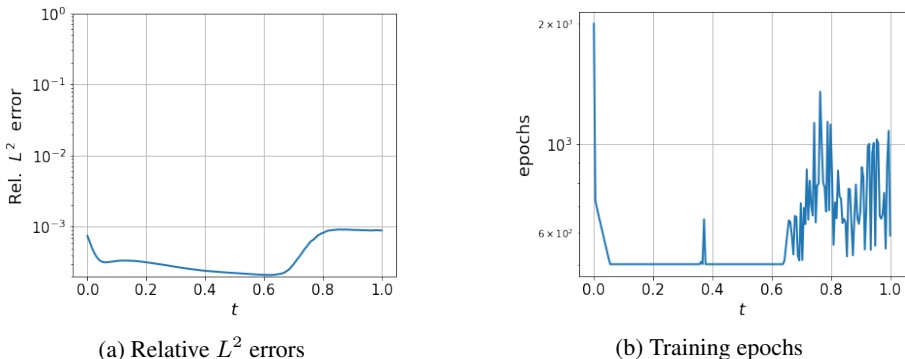

(a) Relative $L^2$ errors          (b) Training epochs

Figure 3: Training results for the Allen-Cahn equation.

## 5.2 ALLEN-CAHN EQUATION

We consider the one-dimensional Allen-Cahn (AC) equation, a benchmark problem for PINN training L. Wight & Zhao (2021); Mattey & Ghosh (2022); Wang et al. (2022a):

$$
\begin{cases}
u_t = \gamma_1 u_{xx} + \gamma_2 u(1 - u^2), & t \in [0, 1], x \in [-1, 1], \\
u(x, 0) = u_0(x), \\
u(t, -1) = u(t, 1), & u_x(t, -1) = u_x(t, 1).
\end{cases} \tag{8}
$$

where $\gamma_1 = 0.0001$, $\gamma_2 = 5$ and $u_0(x) = x^2 \cos(\pi x)$. For the original PINN, the Allen-Cahn equation is hard to solve, but our approach performs well in accuracy and training efficiency. Figure 1 compares the predicted solution to the reference solution. Our technique achieves a relative $L^2$ error of $5.92e - 04$. Figure 3 shows how the $L^2$ error evolves and how many training epochs are needed at different timestamps. The $L^2$ error increases as the AC equation develops more complicated. Each timestamp's training epoch is small across the time domain, reducing training time. More computational results of the AC equation are provided in Appendix A.4.4.

## 5.3 KURAMOTO–SIVASHINSKY EQUATION

The Kuramoto-Sivashinsky (KS) equation is used to model the diffusive–thermal instabilities in a laminar flame front. Existing PINN frameworks are challenging to solve the KS equation as the solution exhibits fast transit and chaotic behaviors Raissi (2018). The KS equation takes the form

$$
\begin{cases}
u_t + \alpha u u_x + \beta u_{xx} + \gamma u_{xxxx} = 0, \\
u(0, x) = u_0(x),
\end{cases} \tag{9}
$$

with periodic boundary conditions. Here $\alpha = 5$, $\beta = 0.5$, $\gamma = 0.005$, and the initial condition $u_0(x) = -\sin(\pi x)$. Figure 4 visualizes the predicted solution against the reference solution, and our proposed method achieves a relative $L^2$ error of $7.17e - 03$. From $t = 0.4$, the reference solution begins to quickly transition, and our method is able to capture this feature. More computational results of the KS equation are provided in Appendix A.4.5.

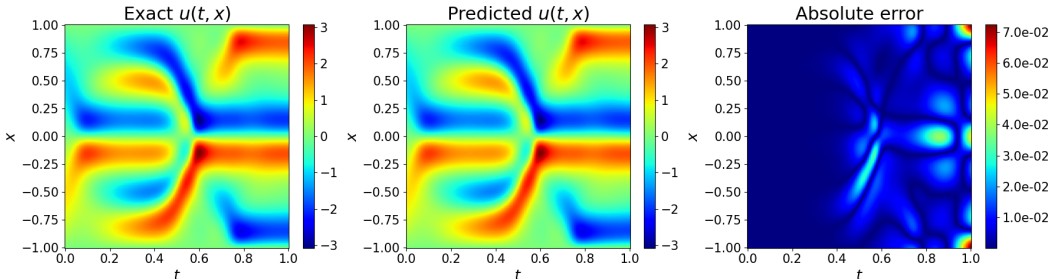

Figure 4: Comparison between the reference and predicted solutions for the Kuramoto–Sivashinsky equation, and the $L^2$ error is $7.17e - 03$.

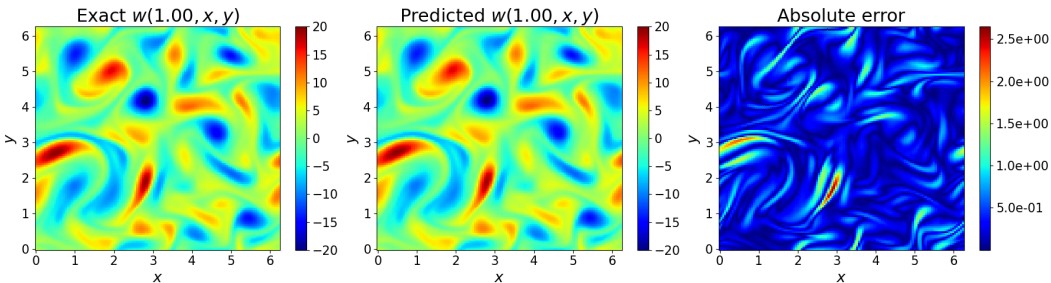

Figure 5: Comparison between the reference and predicted solutions of $w(t, x, y)$ for the Navier-Stokes equation at $t = 1.0$, and the $L^2$ error is $3.44e - 02$.

## 5.4 NAVIER-STOKES EQUATION

We consider the 2D Navier-Stokes (NS) equation in the velocity-vorticity form Wang et al. (2022a)

$$\begin{cases} w_t + \boldsymbol{u} \cdot \nabla w = \frac{1}{\text{Re}} \Delta w, & \text{in } [0, \text{T}] \times \Omega, \\ \nabla \cdot \boldsymbol{u} = 0, & \text{in } [0, \text{T}] \times \Omega, \\ w(0, x, y) = w_0(x, y), & \text{in } \Omega. \end{cases} \tag{10}$$

with periodic boundary conditions. Here, $\mathbf{u} = (u, v)$ represents the flow velocity field, $w = \nabla \times u$ represents the vorticity, and Re is the Reynolds number. In addition, $\Omega$ is set to $[0, 2\pi]^2$ and Re is set to 100. Figure 5 presents the predicted solution of $w(t, x, y)$ compared to the reference solution. Our proposed method can obtain a result similar to that in Wang et al. (2022a), while the training time is only 1/58 of their method. More computational results of the NS equation are provided in Appendix A.4.6.

## 5.5 ABLATION STUDY

We conduct ablation studies on the relatively simpler RD Eq. and AC Eq. to ablate the main designs in our algorithm.

**Time differencing scheme study.** Numerous time differencing schemes have been developed in the last decades. We list some commonly used schemes in Appendix A.2. We do experiments on different time differencing schemes to validate that implicit time differencing schemes (2nd Crank-Nicolson or 4th Gauss-Legendre) are more stable and lead to better performance. The results are given in Table 2.

**Transfer learning study.** To see weather the transfer learning part is effective, we do ablation studies without using transfer learning. Besides, since our strategy of transfer learning is to fine tune all the network parameters, we also do experiments to fine tune the last 1/2/3 layers of the network. The results are given in Table 3. We can see that transfer learning is effective both in the efficiency and accuracy of our method.

Table 2: Time differencing scheme study

| Method | RD Eq. | | AC Eq. | |
|---|---|---|---|---|
| | $L^2$ error | time | $L^2$ error | time |
| Forward Euler | 1.32e-03 | 208 | 9.57e-03 | 304 |
| Backward Euler | 2.74e-03 | 206 | 1.64e-02 | 444 |
| 2nd RK | 1.97e-03 | 761 | 1.17e-03 | 1054 |
| 4th RK | 2.11e-03 | 1187 | 1.31e-03 | 1779 |
| TL-DPINN1 | **1.82e-05** | 1463 | **5.92e-04** | 2328 |
| TL-DPINN2 | 9.34e-05 | 748 | 9.82e-04 | 1100 |

Table 3: Transfer learning study

| Method | RD Eq. | | AC Eq. | |
|---|---|---|---|---|
| | $L^2$ error | time | $L^2$ error | time |
| Without TL | 4.01e-04 | 5880 | 1.35e-02 | 9170 |
| last layer | 3.31e-04 | 638 | 1.01e-02 | 3624 |
| last 2 layers | 3.22e-04 | 221 | 1.01e-02 | 4029 |
| last 3 layers | 4.08e-04 | 232 | 1.01e-02 | 4685 |
| TL-DPINN1 | **1.82e-05** | 1463 | **5.92e-04** | 2328 |
| TL-DPINN2 | 9.34e-05 | 748 | 9.82e-04 | 1100 |

**Repeated test.** To further demonstrate the well-performance of our TL-DPINN method through accuracy and efficiency, we do 5 random runs for RD and AC Eq. by casual PINN and our method for comparison. The results are given in Table 4.

Table 4: Repeated test.

| Method | RD Eq. | | AC Eq. | |
|---|---|---|---|---|
| | $L^2$ error | time | $L^2$ error | time |
| Causal PINN | 3.73e-05 ± 4.66e-06 | 7207 ± 219 | 1.51e-03 ± 2.12e-04 | 9060 ± 341 |
| TL-DPINN1 | **1.76e-05 ± 1.06e-06** | 1463 ± 53 | **6.08e-04 ± 3.06e-05** | 2328 ± 89 |
| TL-DPINN2 | 9.89e-05 ± 8.94e-06 | 811 ± 122 | 9.29e-04 ± 8.06e-05 | 1291 ± 178 |

## 5.6 TRAINING EFFICIENCY

Table 5 illustrates how the computation efficiency is affected by different time discretization methods on different equations. In addition, the casual PINN method is also compared. All neural networks are trained on an NVIDIA GeForce RTX 3080 Ti graphics card. We note that the total training epochs of our methods are not fixed due to the stopping criterion (see Algorithm 1). The training efficiency in Table 5 is consistent with the training time in Table 1.

Table 5: A comparison of training efficiency for different equations.

| Method | Training efficiency (epochs/sec.) | | | |
|---|---|---|---|---|
| | Reaction-Diffusion | Allen-Cahn | Kuramoto-Sivashinsky | Navier-Stokes |
| Casual PINN | 61.70 | 52.33 | 26.24 | 2.77 |
| TL-DPINN1 | **439.37** | **384.47** | **259.20** | **8.32** |
| TL-DPINN2 | 276.40 | 239.52 | 127.55 | 6.37 |

## 6 CONCLUSION

In this paper, we propose a method for solving evolutionary partial differential equations via transfer-learning enhanced discrete physics-informed neural networks (TL-DPINN). The discrete PINNs were thought to be time-consuming and seldom applied in the PINNs literature. We contribute to the PINN community by rediscovering the good performance of the discrete PINNs applied to solve evolutionary PDEs, both theoretically and numerically. Our method first employs a classical numerical implicit time differencing scheme to produce a series of stable propagation equations in space, and then applies PINN approximation to sequentially solve. Transfer learning is used to reduce computational costs while maintaining precision. We demonstrate the convergence property, accuracy, and computational effectiveness of our TL-DPINN method both theoretically and numerically. Our proposed method achieves state-of-the-art results among different PINN frameworks while significantly reducing the computational cost.

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

# A  APPENDIX

## A.1  TABLE OF NOTATIONS

A table of notations is given in Table 1.

## A.2  TIME DIFFERENCING SCHEMES

### A.2.1  EXPLICIT SCHEMES

First-order forward Euler scheme:

$$\frac{u^{n+1}(x) - u^n(x)}{\Delta t} = \mathcal{N}\left[u^n(x)\right]. \tag{11}$$

Second-order explicit Runge-Kutta (2nd RK) scheme:

$$\frac{u^{n+1}(x) - u^n(x)}{\Delta t} = \mathcal{N}\left[u^n(x) + \frac{\Delta t}{2}\mathcal{N}[u^n(x)]\right]. \tag{12}$$

Fouth-order explicit Runge-Kutta (4th RK) scheme:

$$\frac{u^{n+1}(x) - u^n(x)}{\Delta t} = \frac{1}{6}\left[k_1(x) + 2k_2(x) + 2k_3(x) + k_4(x)\right], \tag{13}$$

$$k_1(x) = \mathcal{N}[u^n(x)], \tag{14}$$

$$k_2(x) = \mathcal{N}\left[u^n(x) + \frac{\Delta t}{2}\mathcal{N}[k_1(x)]\right], \tag{15}$$

$$k_3(x) = \mathcal{N}\left[u^n(x) + \frac{\Delta t}{2}\mathcal{N}[k_2(x)]\right], \tag{16}$$

$$k_4(x) = \mathcal{N}\left[u^n(x) + \Delta t\mathcal{N}[k_3(x)]\right]. \tag{17}$$

Table 6: Table of notations

| Notation | Meaning |
|---|---|
| PINN | Physics-informed neural network |
| PDE | Partial differential equation |
| TL-DPINN | Transfer learning enhanced discrete PINN |
| TL-DPINN1 | Crank-Nicolson time differencing in TL-DPINN |
| TL-DPINN1 | Gauss-Legendre time differencing in TL-DPINN |
| $\mathcal{L}$ or $\mathcal{L}^n$ | Physics-informed loss function |
| $\mathcal{N}$ | Differential operator, such as $\mathcal{N}(u) = u_{xx}$ |
| $\mathcal{R}$ | The residual term of the evolutionary PDE, for example $\mathcal{R}(u) = u_t - u_{xx}$ |
| $\Omega$ | Spatial domain |
| $\partial\Omega$ | The boundary of the spatial domain |
| $T$ | End time |
| $N_t$ | Timestamps number |
| $N_b$ | The collocation points number on $\partial\Omega$ |
| $N_u, N_r$ | The collocation points number in $\Omega$ or $\Omega \times [0, T]$ |
| $u(t, x)$ | The exact solution to the evolutionary PDE |
| $u^n(x)$ | The time differencing scheme solution to the evolutionary PDE |
| $u_{\theta^n}(x)$ | The discrete PINN solution to the evolutionary PDE |
| $h_j$ | The $j$ component in the output of the last hidden layer of the neural network |
| $x$ , $x_r$ , $x_b$ | Spatial coordinate |
| $t$ or $t_n$ | Temporal coordinate |
| $\theta$ or $\theta^n, W^n, w^n$ | Neural network parameters |
| $\Delta t$ or $\tau$ | Time step, the interval time between two adjacent timestamps |
| $M_0, M_1$ | Number of maximum iterations in different training stages |
| $\eta$ | The learning rate in gradient descent methods |
| $\epsilon$ | The threshold value |
| $\|\cdot\|$ | The $L^2$ norm of a function, defined by $\|f\| = \left(\int_\Omega |f(x)|^2 dx\right)^{\frac{1}{2}}$ |

### A.2.2 IMPLICIT SCHEMES

First-order backward Euler scheme:

$$\frac{u^{n+1}(x) - u^n(x)}{\Delta t} = \mathcal{N}\left[u^{n+1}(x)\right]. \tag{18}$$

Second-order Trapezoidal scheme:

$$\frac{u^{n+1}(x) - u^n(x)}{\Delta t} = \frac{\mathcal{N}[u^{n+1}(x)] + \mathcal{N}[u^{n+}(x)]}{2}. \tag{19}$$

Second-order Crank-Nicolson scheme (used in TL-DPINN1):

$$\frac{u^{n+1}(x) - u^n(x)}{\Delta t} = \mathcal{N}\left[\frac{u^{n+1}(x) + u^{n+}(x)}{2}\right]. \tag{20}$$

Forth-order Gauss-Legendre scheme (used in TL-DPINN2):

$$\frac{u^{n+1}(x) - u^n(x)}{\Delta t} = \frac{k_1(x) + k_2(x)}{2}, \tag{21}$$

$$k_1(x) = \mathcal{N}\left[u^n(x) + \frac{1}{4}\Delta t k_1(x) + \left(\frac{1}{4} + \frac{\sqrt{3}}{6}\right)\Delta t k_2(x)\right], \tag{22}$$

$$k_2(x) = \mathcal{N}\left[u^n(x) + \left(\frac{1}{4} - \frac{\sqrt{3}}{6}\right)\Delta t k_1(x) + \frac{1}{4}\Delta t k_2(x).\right] \tag{23}$$

The general form of Runge–Kutta schemes with $q$ stages:

$$\frac{u^{n+1}(x) - u^n(x)}{\Delta t} = \sum_{i=1}^{q} b_i k_i(x), \tag{24}$$

$$k_i(x) = \mathcal{N}\left[u^n(x) + \Delta t \sum_{j=1}^{q} a_{ij} k_j(x)\right], \ i = 1, ..., q. \tag{25}$$

where the coefficients $\{a_{ij}, b_i\}$ are determined. Since there are no significant differences for PINN approximation of explicit schemes (i.e. $a_{ij} = 0$ for all $j \geq i$) and implicit schemes (i.e. not all $a_{ij} = 0$ for $j \geq i$), we prefer implicit schemes as they possess the A-stable property to make the time-marching process stable Butcher (2007).

### A.3 THEORETICAL ANALYSIS

### A.3.1 PROOF OF THEOREM 4.1

*Proof.* We split the error $e^n(x) = u(t_n, x) - u_{\theta^n}(x)$ into two parts:

$$e^n(x) = \underbrace{u(t_n, x) - u^n(x)}_{\hat{=}\xi^n(x)} + \underbrace{u^n(x) - u_{\theta^n}(x)}_{\hat{=}\eta^n(x)} \tag{26}$$

The first term $\xi^n(x)$ estimates the error from the Crank-Nicolson time differencing schemes. From Lemma A.1 we have $\|\xi^n\| \leq C\tau^2$. The second term $\eta(x)$ estimates the error from the PINN approximation in space and the cumulative effect of time. From Lemma A.2 we have $\|\eta^n\| \leq C\sqrt{t_n}(\max_{1 \leq i \leq n} \sqrt{\mathcal{L}^i} + N_r^{\frac{1}{4}})$. Then by the triangular inequality, we finish the proof. □

### A.3.2 SOME LEMMAS IN THE PROOF OF THEOREM 4.1

**Lemma A.1.** *Denote $\xi^n(x) = u(t_n, x) - u^n(x)$, where $u(t_n, x)$ is the exact solution to evolutionary PDEs and $u^n(x)$ is the Crank-Nicolson time differencing discrete solution, then we have the estimate*

$$\|\xi^n\| \leq C\tau^2, \tag{27}$$

*for some constant $C$ independent of time step $\tau$, collocation points number $N_r$ and trained loss value $\mathcal{L}^n$.*

*Proof.* Firstly, we replace $u^n(x)$ in the Crank-Nicolson time differencing scheme by the evolutionary PDE's solution $u(t_n, x)$ and compare the difference. This can be achieved by the standard Taylor expansion techniques. We do Taylor expansion at the point $t_{n+\frac{1}{2}} = (n + \frac{1}{2})\tau$ to get

$$\frac{u(t_{n+1}, x) - u(t_n, x)}{\tau} = u_t(t_{n+\frac{1}{2}}, x) + \mathcal{O}(\tau^2),$$

and

$$\mathcal{N}\left[\frac{u(t_{n+1}, x) + u(t_n, x)}{2}\right] = \mathcal{N}\left[u(t_{n+\frac{1}{2}}, x)\right] + \mathcal{O}(\tau^2).$$

Noticing that $u(t, x)$ is satisfied with the evolutionary PDE $u_t = \mathcal{N}[u]$, we have

$$\frac{u(t_{n+1}, x) - u(t_n, x)}{\tau} = \mathcal{N}\left[\frac{u(t_{n+1}, x) + u(t_n, x)}{2}\right] + \mathcal{O}(\tau^2). \tag{28}$$

Now subtracting equation 28 from the Crank-Nicolson scheme, we obtain the relation of the propagation error $\xi^n(x) = u(t_n, x) - u^n(x)$ as

$$\frac{\xi^{n+1}(x) - \xi^n(x)}{\tau} = \mathcal{N}\left[\frac{u(t_{n+1}, x) + u(t_n, x)}{2}\right] - \mathcal{N}\left[\frac{u^{n+1}(x) + u^n(x)}{2}\right] + \mathcal{O}(\tau^2), \tag{29}$$

Secondly, we estimate the $L^2$ norm error estimate of $\xi^n(x)$. This can be achieved by the standard Höder inequality estimate techniques. We multiply equation 29 by $\frac{1}{2}(\xi^{n+1}(x) + \xi^n(x))$ and integrate for $x$ on the domain $\Omega$. With Assumption 4.1 holds, we have

$$\frac{\left\|\xi^{n+1}\right\|^2 - \left\|\xi^n\right\|^2}{2\tau} \leq \int_\Omega \mathcal{O}(\tau^2) \cdot \frac{\xi^{n+1}(x) + \xi^n(x)}{2}$$

$$\leq C_0 \tau^4 + \frac{1}{2}\left\|\xi^{n+1}\right\|^2 + \frac{1}{2}\left\|\xi^n\right\|^2,$$

for some constant $C_0$ only depends on $u(t, x)$ and its derivatives. We rearrange it to the following form

$$\left\|\xi^{n+1}\right\|^2 \leq \frac{1+\tau}{1-\tau}\left\|\xi^n\right\|^2 + \frac{2C_0}{1-\tau}\tau^5.$$

Since $\xi^0(x) = 0$, we apply Lemma A.3 to get

$$\left\|\xi^n\right\|^2 \leq \frac{2C_0\tau^5}{1-\tau} \cdot \frac{\left(\frac{1+\tau}{1-\tau}\right)^n - 1}{\frac{1+\tau}{1-\tau} - 1}$$

$$\leq 6C_0 t_n \tau^4.$$

So we have $\|\xi^n\| \leq C\sqrt{t_n}\tau^2$ for some constant $C = \sqrt{6C_0}$ and we finish the proof. $\qquad\square$

**Lemma A.2.** *Denote $\eta^n(x) = u^n(x) - u_{\theta^n}(x)$, where $u^n(x)$ is the Crank-Nicolson time differencing discrete solution and $u_{\theta^n}(x)$ is the discrete PINN solution, then we have the estimate*

$$\|\eta^n\| \leq C\sqrt{t_n}(\max_{1 \leq i \leq n} \sqrt{\mathcal{L}^i} + N_r^{\frac{1}{4}}), \tag{30}$$

*Proof.* The PINN solution $u_{\theta^{n+1}}(x)$ is obtained by optimize the physics-informed loss $\mathcal{L}^{n+1}(\theta^{n+1})$. Define the residual function $\mathcal{R}^{n+1}(x)$ by

$$\mathcal{R}^{n+1}(x) = \frac{u_{\theta^{n+1}}(x) - u_{\theta^n}(x)}{\tau} - \mathcal{N}\left[\frac{u_{\theta^{n+1}}(x) + u_{\theta^n}(x)}{2}\right], \quad \forall x \in \Omega. \tag{31}$$

The loss $\mathcal{L}^{n+1}(\theta^{n+1})$ is partially composed of the residual function on some randomly sampled point, so

$$\mathcal{L}^{n+1} \geq \frac{\lambda_r}{N_r}\sum_{i=1}^{N_r}|\mathcal{R}(x_r^i)|^2.$$

By the Monte-Carlo quadrature rule in the numerical integration method, we can estimate the $L^2$ norm of the residual function $\mathcal{R}(x)$ by the discrete form

$$\left\|\mathcal{R}^{n+1}\right\|^2 = \int_\Omega |\mathcal{R}^{n+1}(x)|^2 dx$$

$$\leq \frac{1}{N_r} \sum_{i=1}^{N_r} |\mathcal{R}(x_r^i)|^2 + C_1 N_r^{-\frac{1}{2}}$$

$$\leq \frac{\mathcal{L}^{n+1}}{\lambda_r} + C_1 N_r^{-\frac{1}{2}},$$

for some constant $C_1$ depends on the regularities of the PINN solution $u_{\theta^n}(x)$.

Now we turn to estimate the $L^2$ norm error estimate of $\eta^n(x)$. We first replace $u^n(x)$ in the Crank-Nicolson time differencing scheme by the PINN solution $u_{\theta^n}(x)$ and compare the difference. Subtracting equation 31 from the Crank-Nicolson scheme, we obtain the relation of the propagation error $\eta^n(x) = u^n(x) - u_{\theta^n}(x)$ as

$$\frac{\eta^{n+1} - \eta^n}{\tau} - \left(\mathcal{N}\left[\frac{u^{n+1}(x) + u^n(x)}{2}\right] - \mathcal{N}\left[\frac{u_{\theta^{n+1}}(x) + u_{\theta^n}(x)}{2}\right]\right) = -\mathcal{R}(x) \qquad (32)$$

Similar to the proof in Lemma A.1, we multiply equation 32 by $\frac{1}{2}(\eta^{n+1}(x) + \eta^n(x))$ and integrate for $x$ on the domain $\Omega$. With Assumption 4.1 holds, we have

$$\frac{\left\|\eta^{n+1}\right\|^2 - \left\|\eta^n\right\|^2}{2\tau} \leq -\int_\Omega \mathcal{R}(x) \cdot \frac{\eta^{n+1}(x) + \eta^n(x)}{2}$$

$$\leq \frac{1}{4}\left\|\mathcal{R}^{n+1}\right\|^2 + \frac{1}{2}\left\|\eta^{n+1}\right\|^2 + \frac{1}{2}\left\|\eta^n\right\|^2,$$

then we rearrange it to the following form

$$\left\|\eta^{n+1}\right\|^2 \leq \frac{1+\tau}{1-\tau}\left\|\eta^n\right\|^2 + \frac{\tau}{1-\tau}\left\|\mathcal{R}^{n+1}\right\|^2.$$

then we apply Lemma A.3 to get

$$\left\|\eta^n\right\|^2 \leq \left(\frac{1+\tau}{1-\tau}\right)^n \left\|\eta^0\right\|^2 + \frac{\left(\frac{1+\tau}{1-\tau}\right)^n - 1}{\frac{1+\tau}{1-\tau} - 1} \cdot \frac{\tau \max\limits_{1 \leq i \leq n} \left\|\mathcal{R}^i\right\|^2}{1-\tau}$$

$$\leq (1 + 6t_n)\left\|\eta^0\right\|^2 + \frac{3t_n}{2} \max\limits_{1 \leq i \leq n} \left\|\mathcal{R}^i\right\|^2.$$

Since $\eta^0(x) = 0$, we have $\|\eta^n\| \leq C\sqrt{t_n}(\max\limits_{1 \leq i \leq n} \sqrt{\mathcal{L}^i} + N_r^{\frac{1}{4}})$ for some constant $C$ and we finish the proof. $\qquad \square$

**Lemma A.3.** *If the sequence $\{T_n\}_{n=0}^\infty$ satisfies the following propagation relation for some positive constant $\alpha$ and $\{\beta_n\}_{n=1}^\infty$:*

$$T_{n+1} \leq \alpha T_n + \beta_{n+1}, \quad n \geq 0,$$

*then we have*

$$T_n \leq \alpha^n T_0 + \frac{\alpha^n - 1}{\alpha - 1} \max\limits_{1 \leq i \leq n} \beta_i, \quad n \geq 1.$$

*Proof.* This is accomplished by a standard recurrence formula. $\qquad \square$

## A.4 EXPERIMENTAL DETAILS

In this section, we provide the details on the numerical experiments of Section 5.

### A.4.1 Neural Network Architecture

We present two practical considerations for the PINN network architecture, which has been applied in CausualPINN Wang et al. (2022a) and other PINN frameworks. Although not deemed crucial for the successful application of Algorithm 1, we have empirically observed that including them can lead to further enhancements in accuracy and computational efficiency.

**Fourier Features Embedding.** Many researchers have utilized Fourier features embedding to enhance the accuracy and generalization Tancik et al. (2020); Wang et al. (2021b). We employ 1-D Fourier features embedding in the following format:

$$\gamma(x) = [1, \cos(\omega x), \sin(\omega x), \cos(2\omega x), \sin(2\omega x), ..., \cos(M\omega x), \sin(M\omega x)]^T$$

where $\omega = 2\pi/L$ and $M$ is a positive integer hyper-parameter. It maps the input data to a higher dimensional space by Fourier transforms. The major advantage of this technique is that it improves the model's ability to approximate periodic or oscillatory behavior in the input data. It allows us to satisfy the periodic boundary condition as

$$g(x_b^i) = g(x_b^i + L)$$

where $L$ represents the period of the periodic boundary condition. Furthermore, for the two-dimensional Navier-Stokes equation, the Fourier feature embedding takes the following form

$$\gamma(x) = \begin{bmatrix} 1 \\ \cos(\omega_x x), ..., \cos(M\omega_x x) \\ \cos(\omega_y y), ..., \cos(M\omega_y y) \\ \sin(\omega_x x), ..., \sin(M\omega_x x) \\ \sin(\omega_y y), ..., \sin(M\omega_y y) \\ \cos(\omega_x x)\cos(\omega_y y), ..., \cos(M\omega_x x)\cos(M\omega_y y) \\ \cos(\omega_x x)\sin(\omega_y y), ..., \cos(M\omega_x x)\sin(M\omega_y y) \\ \sin(\omega_x x)\cos(\omega_y y), ..., \sin(M\omega_x x)\cos(M\omega_y y) \\ \sin(\omega_x x)\sin(\omega_y y), ..., \sin(M\omega_x x)\sin(M\omega_y y) \end{bmatrix}$$

Previous studies Lu et al. (2021b); Sukumar & Srivastava (2022) have shown that this method can generally be applied to any problem that exhibits periodic or oscillatory behavior, regardless of the particular boundary conditions involved. For instance, Fourier feature embedding can be employed to solve problems with Dirichlet boundary conditions in which the solution is specified at the boundary (or Neumann boundary conditions in which the solution's derivative is specified at the boundary). In such a scenario, the embedding technique can be used to capture the periodic and oscillatory behavior of the input data, while the neural network can be trained to satisfy the Dirichlet boundary conditions (or Neumann boundary conditions).

**Modified Multi-layer Perceptrons.** In recent researches Wang et al. (2022a; 2021a), "modified MLP", a novel multi-layer perceptron architecture, has been proposed. Compared to conventional multi-layer perceptrons, the "modified MLP" demonstrates superior performance because it excels at capturing steep gradients and minimizing residuals of partial differential equations. The form of this architecture is given as:

$$\begin{cases} U = \sigma(XW_u + b_u), \\ V = \sigma(XW_v + b_v), \\ H_{(1)} = \sigma(XW_{(0)} + b_{(0)}), \\ Z_{(n)} = \sigma(H_{(n)}W_{(n)} + b_{(n)}), \quad n = 1, 2, ..., D-1. \\ H_{(n+1)} = (1 - Z_{(n)}) \odot U + Z_{(n)} \odot V, \quad n = 1, 2, ..., D-1. \\ u_\theta(X) = H_{(D)}W_{(D)} + b_{(D)}. \end{cases} \tag{33}$$

where $\sigma(\cdot)$ represents activation function ($\tanh(\cdot)$ in this work); the trainable parameters of the neural network are indicated by $W_u, W_v, W_{(n)}, b_u, b_v, b_{(n)}$; $D$ represents the depth of neural network; and $\odot$ denotes the operation of point-wise multiplication. The use of skip connections or residual connections is a significant distinction between "modified MLP" and conventional MLP. These connections enable the network to bypass certain layers and transmit information directly from earlier layers to later layers.

**Multiple Neural Networks.** For PINN with backward Euler or Crank-Nicolson time differencing, the neural network has the form of single input $x$ and single output $u_\theta(x)$. However, for the general form of Runge-Kutta with $q$ stages, we have multiple outputs $[k_1(x), k_2(x), \cdots, k_q(x), u^{n+1}(x)]$. While it is possible to use a single neural network with multiple outputs for the PINN approximation, this approach may lead to slow convergence. This is because the hidden function $k_i(x)$ can differ in scale from the solution $u^{n+1}(x)$. Instead, we use $q + 1$ neural networks to separately approximate $k_1(x), k_2(x), \cdots, k_q(x), u^{n+1}(x)$. Although this approach leads to an increase in the number of neural network parameters, it greatly enhances both the training efficiency and accuracy.

### A.4.2 Configuration of Training

**Error metric** To quantify the performance of our methods, we apply a relative $L^2$ norm over the spatial-temporal domain:

$$\text{relative } L^2 \text{ error} = \sqrt{\frac{\sum_{n=1}^{N_t} \sum_{i=1}^{N_r} |u_{\theta^n}(x_i) - u(t_n, x_i)|^2}{\sum_{n=1}^{N_t} \sum_{i=1}^{N_r} u(t_n, x_i)^2}} \tag{34}$$

**Neural networks and training parameters** In all examples, the Fourier feature embedding is applied and the modified MLP is used. Multiple neural networks are used in our TL-DPINN2 method while a single neural network is used in our TL-DPINN1 method. Adam optimizer with an initial learning rate of 0.001 and exponential rate decay is used. More details about the hyper-parameters of neural networks and the hyper-parameters of Algorithm 1 are presented in Table 7.

Table 7: Detailed experimental settings of Section 5.

| Equations | Depth | Width | Features $M$ | $N_t$ | $N_r$ | Iterations ($M_0$,$M_1$) | $\epsilon$ |
|---|---|---|---|---|---|---|---|
| RD | 4 | 128 | 10 | 200 | 512 | (10000,1000) | 1e-9 |
| AC | 4 | 128 | 10 | 200 | 512 | (10000,2000) | 1e-10 |
| KS(regular) | 3 | 256 | 5 | 250 | 500 | (10000,3000) | 1e-8 |
| KS(chaotic) | 8 | 128 | 5 | 250 | 500 | (10000,7000) | 1e-10 |
| NS | 4 | 128 | 5 | 100 | 100 | (10000,5000) | 1e-5 |

For the configuration of other five baselines: 1) original PINN Raissi et al. (2019); 2) adaptive sampling L. Wight & Zhao (2021); 3) self-attention McClenny & Braga-Neto (2023); 4) time marching Mattey & Ghosh (2022) and 5) causal PINN Wang et al. (2022a), all of them have a neural network size with the same width and 1 deeper depth than that in Table 7. The collocation points number for all five baselines are configured to be $N_t \times N_r$ in Table 7. For example, a continuous original PINN has size $[2, 128, 128, 128, 128, 128, 1]$ and $200 \times 512$ collocation points on the space-time domain to compute the loss, then each discrete PINN has the smaller size $[1, 128, 128, 128, 128, 1]$ and much smaller collocation points $512$ on space domain. The total parameters and computation of 200 discrete PINNs and the computation on the loss calculation are about the same with a single continuous PINN. In this configuration, we can sure that the comparison between our TL-DPINNs and other five baselines is fair, showing the discrete PINNs are efficient for practical applications.

### A.4.3 Additional results for Reaction-Diffusion equation

Figure 6 (a) depicts how the $L^2$ error changes as time goes on, as we can see, the $L^2$ error increases in the early training steps and is kept at a stable level between $1.00e - 05$ and $5.00e - 05$ later. As shown in Figure 6 (b), based on the trainable parameters of the preceding time stamp, only a few hundred steps of training are required for each time stamp to satisfy the early stopping criterion, and then move to the training of the next time stamp. Figure 8 shows the training loss at different time steps. Figure 7 compares the predicted and reference solutions at different time instants. The predictions given by our method are identical to the reference solutions.

### A.5 Additional results for Allen-Cahn equation

Figure 9 shows the predicted solution against the reference solution, our proposed method achieves a relative $L^2$ error of $5.92e - 04$. Figure 10 presents the comparison between the reference and the

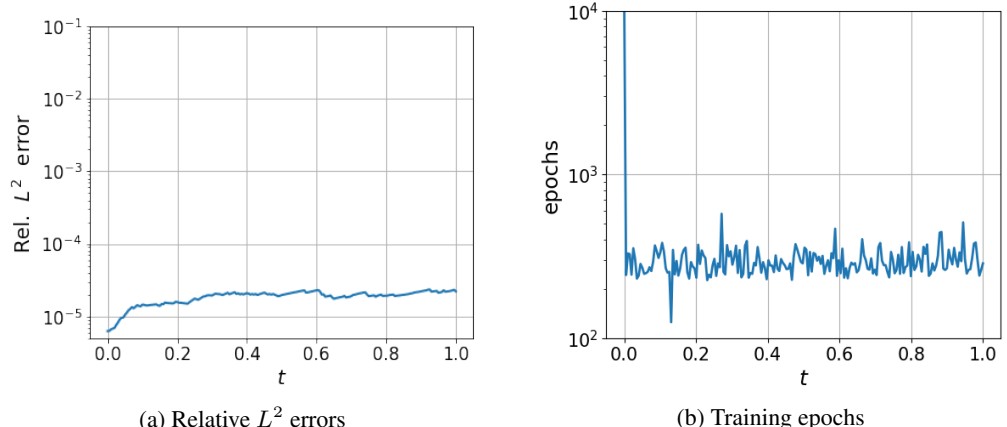

(a) Relative $L^2$ errors

(b) Training epochs

Figure 6: Training results for the Reaction-Diffusion equation.

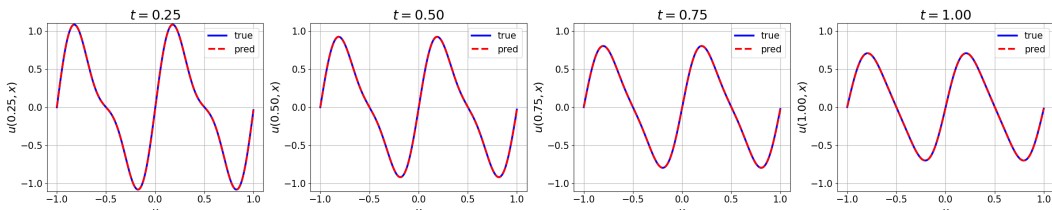

Figure 7: Comparison between the predicted and reference solutions at different time instants for the Reaction-Diffusion equation.

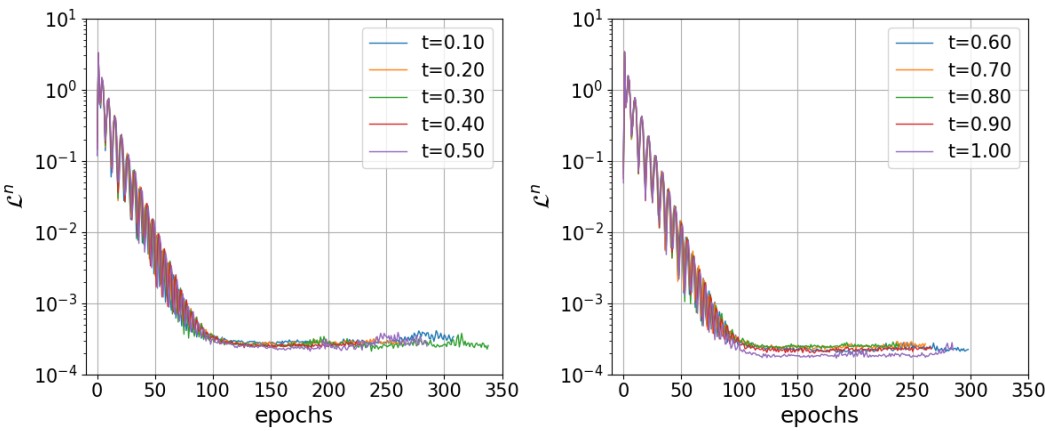

Figure 8: Loss curves at different time steps for the Reaction-Diffusion equation.

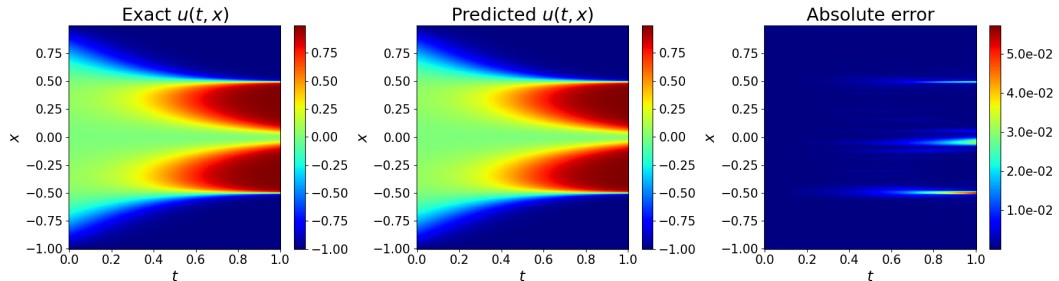

Figure 9: Comparison between the reference and predicted solutions for the Allen-Cahn equation, and the $L^2$ error is $4.04e - 03$.

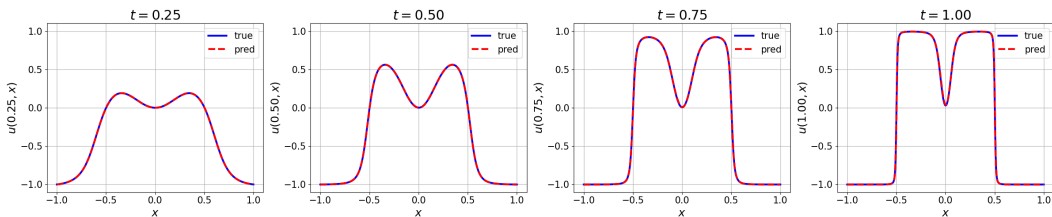

Figure 10: Comparison between the predicted and reference solutions at different time instants for the Allen-Cahn equation.

predicted solutions at given time instants $t = 0.25, 0.50, 0.75, 1.00$. As time goes on, our method is capable of exactly fitting the evolutionary reference solution.

### A.6 ADDITIONAL RESULTS FOR KURAMOTO–SIVASHINSKY EQUATION

**Regular.** The example presented in Section 5.3 shows a relatively regular solution. From Figure 11 (a), we can figure out how the $L^2$ error changes with the evolution of the equation. The $L^2$ error is relatively small in the early time stamps compared with the $L^2$ error in later time stamps for the solution happens to experience a fast transition as time goes on. Figure 11 (b) represents the training epochs required at different time steps. The KS equation tends to become complex at around $t = 0.5$, leading to a drastic surge in demand for training epochs. Figure 12 presents the comparison between the reference and the predicted solutions at different time moments $t = 0.2, 0.4, 0.6, 0.8, 1.0$, and it is clear that our predicted solution is highly consistent with the reference solution.

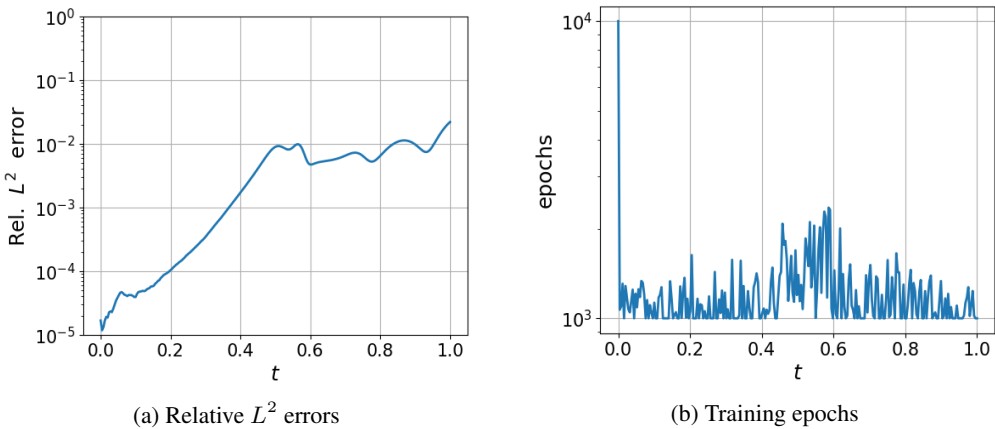

(a) Relative $L^2$ errors

(b) Training epochs

Figure 11: Training results for the Kuramoto–Sivashinsky (regular) equation.

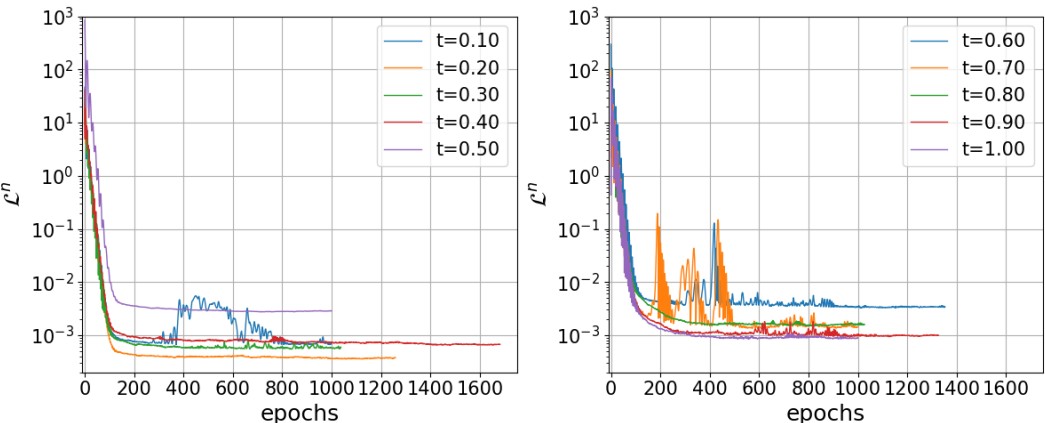

Figure 12: Comparison between the predicted and reference solutions at different time instants for the Kuramoto–Sivashinsky (regular) equation.

Figure 13: Loss curves at different time steps for the Kuramoto–Sivashinsky (regular) equation.

**Chaotic.** We consider using the Kuramoto-Sivashinsky equation to describe more complex chaotic phenomena, in which $\alpha = 100/16$, $\beta = 100/16^2$, $\gamma = 100/16^4$, and the initial condition $u_0(x) = \cos(x)(1 + \sin(x))$. The comparison between the reference and the predicted solution is visualized in Figure 14. As discussed in the previous section, PINN has difficulty learning sharp features for a larger number of evolutionary equations. However, our proposed method can learn solutions to chaotic phenomena. Our proposed method gives a relative $L^2$ error of $3.74e - 01$, whose variation trend is shown in Figure 15 (a). As shown in Figure 15 (b), with the reference solution becoming complex later in the training process, the maximum of the training epoch is always reached.

From a critical standpoint, here we should also mention that difficulties can still arise in simulating the long-time behavior of chaotic systems. We observe that our predicted solution accurately captures the transition to chaos at around $t = 0.4$, while eventually losing accuracy after $t = 0.8$ as depicted in Figure 14, as well as in CasualPINN Wang et al. (2022a). Figure 16 depicts the comparison between the predicted and reference solution at different time instants. From $t = 0.4$, our method has difficulty in fitting the reference solution exactly and the contrast in the final state is even worse. This may be due to the chaotic nature of the problem and the inevitable numerical error accumulation of PINNs, which have appeared and been discussed in Wang et al. (2022a).

### A.6.1 ADDITIONAL RESULTS FOR NAVIER-STOKES EQUATION

Our method is effective in solving NS Eq. with turbulence behavior. As shown in Figure 18, only one thousand training epochs are required on average for each timestamp to converge. Figure 20 shows additional comparisons of $w(t, x, y)$ at different time stamps. As time passes, both the absolute error and the $L^2$ error between the reference and predicted $w(t, x, y)$ increase gradually. Figure 19 shows how the loss value decreases at different timestamps, where $\mathcal{L}_w^n$ is the loss for the equation $w_t + u_{\theta^n} \cdot \nabla w - \frac{1}{\text{Re}} \Delta w = 0$, and $\mathcal{L}_c^n$ for the equation $\nabla \cdot u_{\theta^n} = 0$.

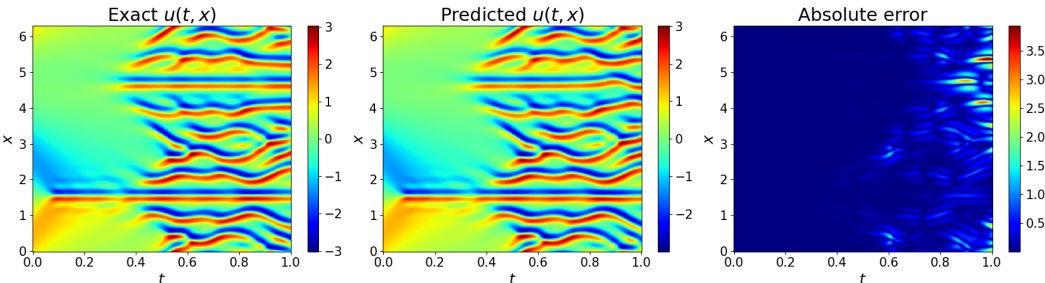

Figure 14: Comparison between the reference and predicted solutions for the Kuramoto–Sivashinsky(chaotic) equation, and the $L^2$ error is $3.74e - 01$.

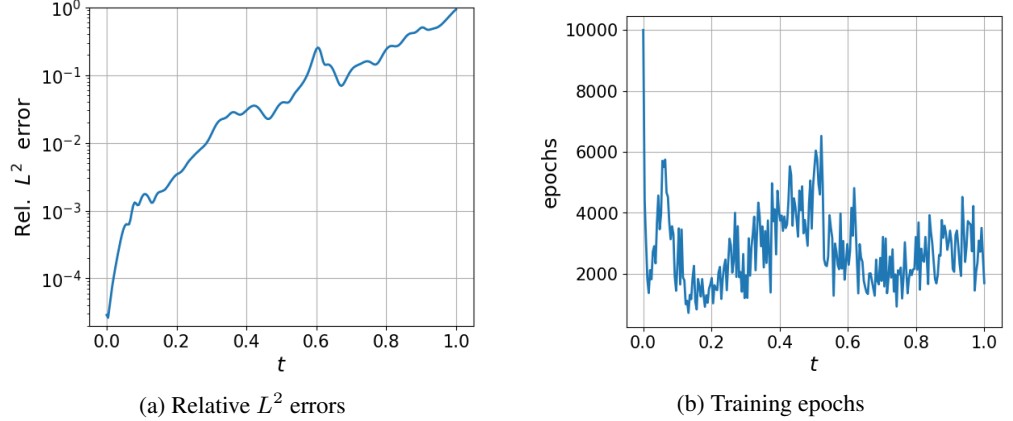

(a) Relative $L^2$ errors

(b) Training epochs

Figure 15: Training results for the Kuramoto–Sivashinsky (chaotic) equation.

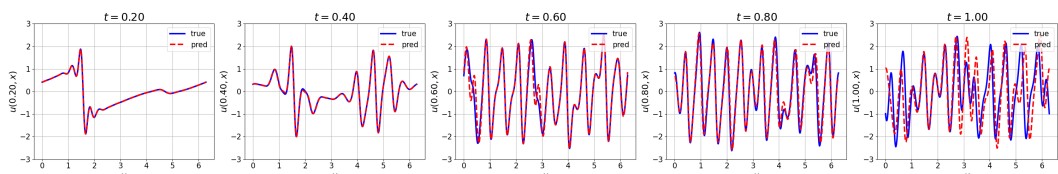

Figure 16: Comparison between the predicted and reference solutions at different time instants for the Kuramoto–Sivashinsky(chaotic) equation.

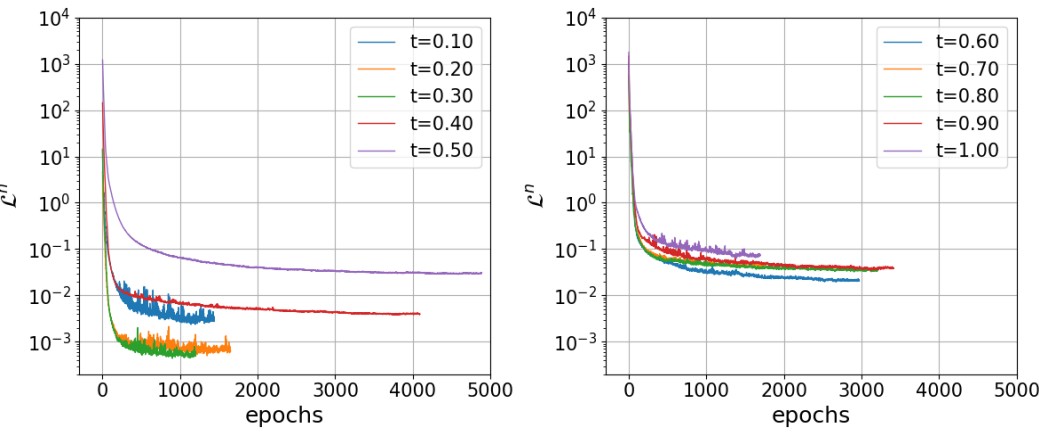

Figure 17: Loss curves at different time steps for the Kuramoto–Sivashinsky (chaotic) equation.

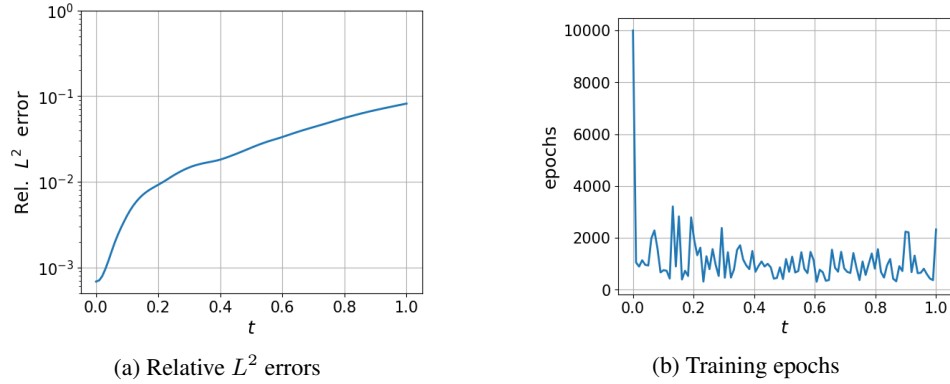

(a) Relative $L^2$ errors

(b) Training epochs

Figure 18: Training results for the Navier-Stokes equation.

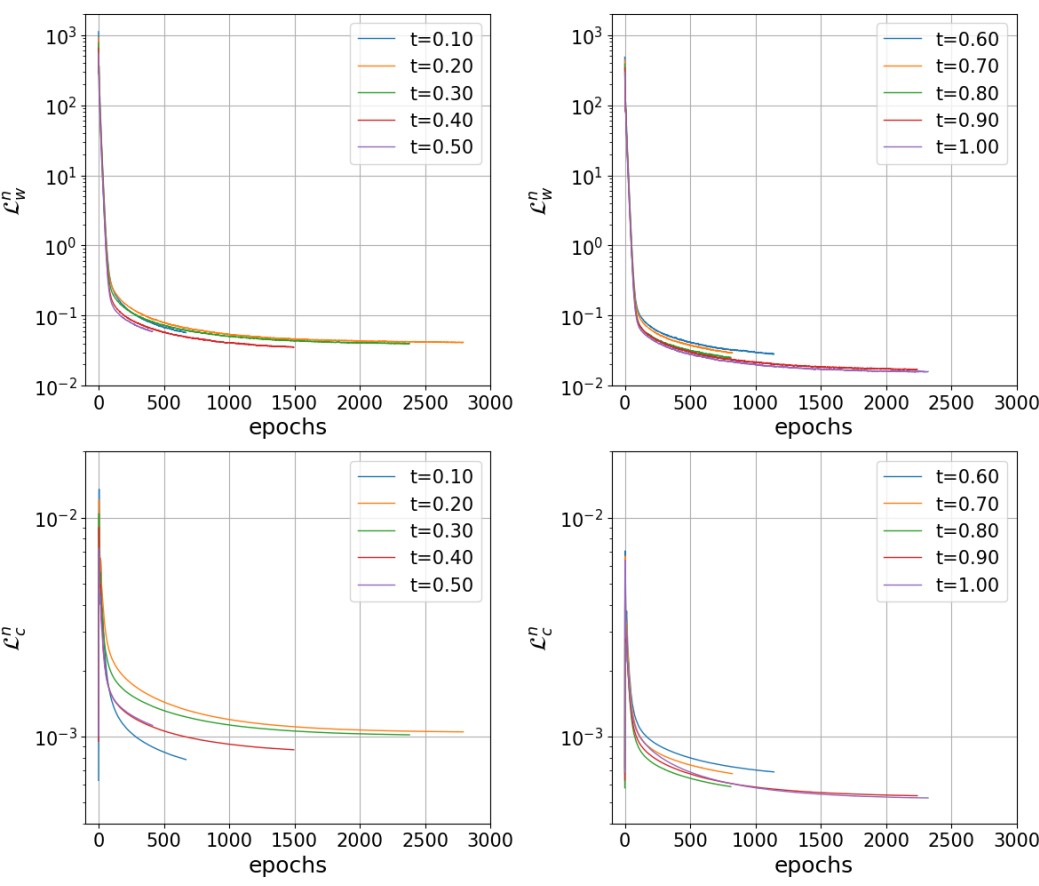

Figure 19: Loss curves at different time steps for the Navier-Stokes equation.

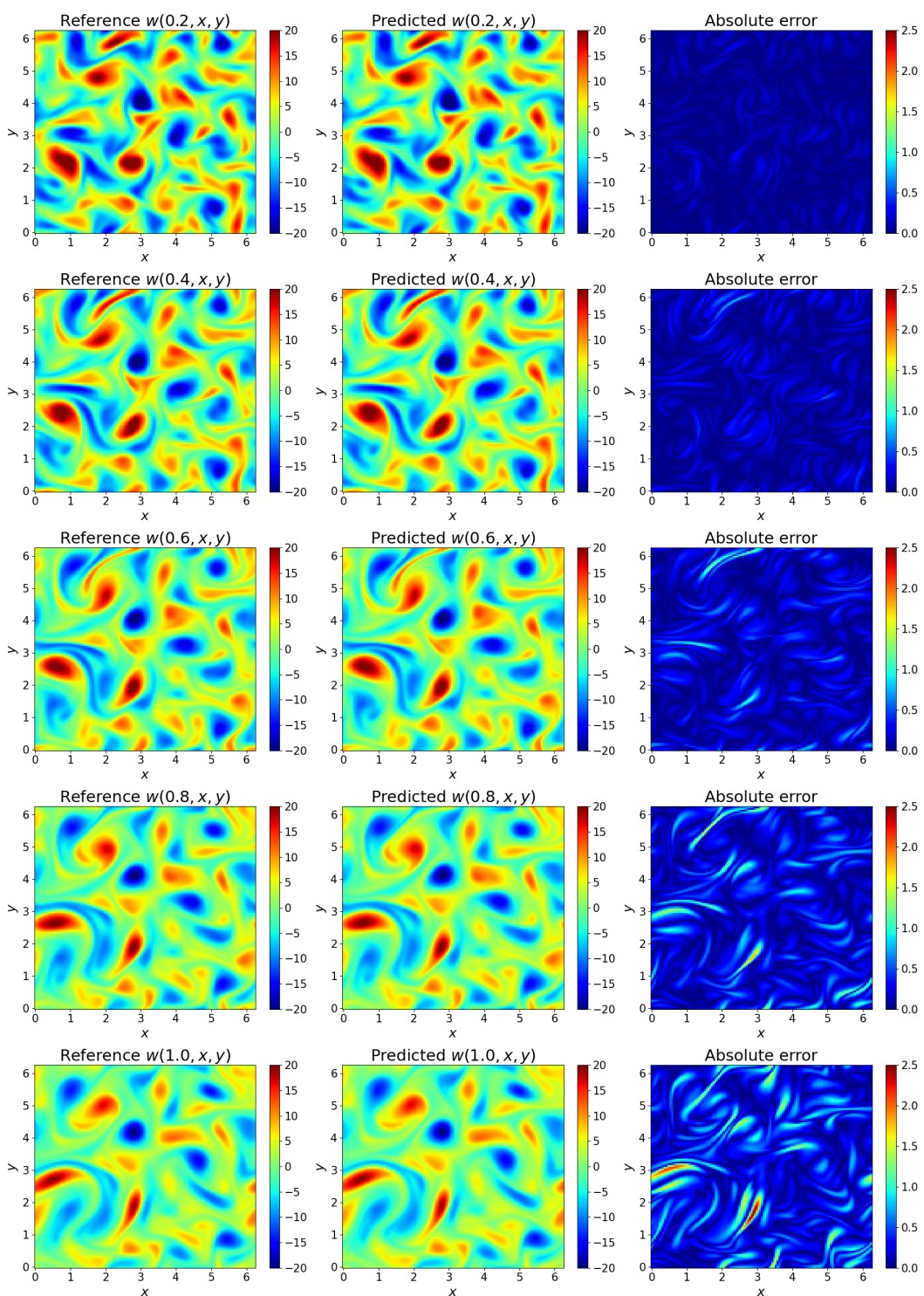

Figure 20: Comparison between the reference and predicted solutions of $w(t, x, y)$ for the Navier-Stokes equation at $t = 0.2, 0.4, 0.6, 0.8, 1.0$.

