# OpenReview forum: "Efficient Discrete Physics-informed Neural Networks for Solving Evolutionary Partial Differential Equations"
_ICLR.cc/2024/Conference — ICLR 2024 Conference Withdrawn Submission_

### Official Review · Reviewer_HW8p · 2023-10-30

**Soundness:** 2 fair
**Presentation:** 1 poor
**Contribution:** 1 poor
**Rating:** 3
**Confidence:** 5

**Summary:**

The paper proposes a discretization method for time-dependent partial differential equations. Time is discretized using a standard Crank-Nicolson scheme. At every time step, this yields a time-dependent PDE that only has spatial derivatives. This time-independent PDE (at each time step) is approximated by Neural networks by minimizing a PINN-type *physics-informed loss function* based on the residual of the time-independent PDE at each time step. Thus, a series of neural networks are trained sequentially in order to approximate the solution. Moreover, the training at each time-step is initialized with weights from the previous time-step constituting a *transfer learning* procedure. A error estimate is provided for this method for a semi-linear PDE with some assumptions on the nonlinearity. Numerical results comparing the method with other PINN baselines is also provided for a set of PDEs in one and two space dimension (Reaction-Diffusion, Allen-Cahn, Kumamoto-Shivashinkshy and Navier-Stokes).

**Strengths:**

- The paper attempts to address an important question: how to accelerate the training and enhance the performance of neural network based PINN type methods for the numerical solution of partial differential equations.

- An error estimate is derived.

- Standardized benchmarks are presented.

**Weaknesses:**

- **Novelty**: It is difficult to see what is truly novel in this paper. The idea of using a time-discrete, also called as method of lines, PINN type formulation where a standard time stepping scheme (implicit RK2) leads to time-independent PDEs, which are then discretized with neural networks with a PINN type loss function, is already explicitly stated in Raissi et al paper reintroducing PINNs (the ideas of PINNs go back to Dissanayake et al in 1994 and Lagaris et al in 1998 and 2000). This fact is also acknowledged by the authors themselves. All the authors have done is to replace Implicit RK2 with Crank-Nicolson. This does not constitute sufficient novelty at all as anyone can replace an existing ODE solver with another existing ODE solver -- there is no reason to believe that CN is any better than implicit RK2. Moreover, what the authors call as transfer learning is pretty straightforward as all the authors do is to initialize the weights at each time step with those at the previous time-step. Although I did not check the code of the Raissi et al paper, I am fairly sure that they do the same too -- what else can make sense ? Moreover, the theoretical result is a straightforward adaptation of well-known techniques (see Mishra and Molinaro) and is only holding for a very strong condition on the nonlinearity (monotonicity of the nonlinear operator) which is not satisfied for any of the numerical examples presented here !! Furthermore, Mishra and Molinaro already proved an error estimate for the original PINNs for time-dependent PDEs -- so theory cannot discriminate between PINNs and discrete PINNs, at least at the level of approximate error, meaning that the theoretical contribution is not novel either.

- **Rationale and Numerical Experiments** Given the above caveats on novelty, a possible strength of the paper could be impressive numerical demonstration. Examining the paper closer on this front, we see that the authors consider 4 fairly standard test cases, 3 of them one-dimensional PDEs (RD, AC and KS) and one, the standard two-dimensional Navier-Stokes equations on a periodic square with a Reynolds number of 100. The baselines are all PINN-based or their variants, including casual PINNs. While the proposed method TL-DPINN1 and TL-DPINN2 is the best-performing, both with respect to accuracy and time over standard PINNs, the results are totally underwhelming to any reader familiar with numerical methods. For instance, it takes 748 Secs or more than 12 minutes to solve a straightforward 1-D reaction-diffusion equation with their best-performing method. What is the time taken (for similar accuracy) with a standard Crank-Nicolson based finite-difference scheme ? I suspect that it will be at best a few seconds. So, your method is two orders of magnitude closer to than text book numerical solvers !! Similarly for the 2-D Navier-Stokes, this reviewer ran a straightforward spectral method solvers to obtain a run-time of less than 0.1 secs for an accuracy of $3\%$ -- even a textbook solver will take around a few secs. On the other hand, your best performing method takes more than 3 hours (is 4 orders of magnitude slower) !! -- so I urge the authors to present some standard numerical method based baseline to see how slow your method is compared to what is out there. The authors will argue that they have baselined against PINNs. This argument is not good enough as for a method to be adopted, it has to be compared to the best that is out there, not just neural network baselines. Otherwise, publication in a leading ML venue is difficult to justify.


Of course, PINNs and their variants might have other advantages that standard numerical methods don't -- for instance problems with complex geometries, very high-dimensional problems (such as Radiative transfer, see Mishra and Molinaro JQSRT for instance) or parametric problems. However, the authors do not consider any of these problems and restrict their attention to simple problems where numerical solver baselines are several orders of magnitude faster, diminishing the impact of their contribution.

**Questions:**

1. Last sentence of Page4 and first sentence of page 5; you say that transfer learning is performed by solely updating the weights in the last hidden layer per time step. However in Algorithm 1 (page 5, line 10) it seems that all the parameters are updated after initializing them from the trained parameter at previous time-step. Which one is it ?

2. Please address the questions that arise in the weaknesses section, particularly about novelty and numerical baselines.

3. How do you choose the time-step for the Crank-Nicolson method ? In particular, can you choose large time-steps for stiff problems. In that case, please demonstrate how the method performs by considering the following wave equation:

$u_t + \frac{a+b}{2} u_x + \frac{b-a}{2} v_x = 0, v_t + \frac{b-a}{2} u_x + \frac{a+b}{2} v_x = 0$. You can use transparent boundary conditions -- the two wave speeds are a and b. Choose a << b to make the problem stiff and see how the time-step can be chosen.

---

> ### Author Response · Authors · 2023-11-15
>
> **W1. Novelty.** The novelty of the paper is that we contribute to the PINN community by rediscovering the good performance of the discrete PINNs applied to solve evolutionary PDEs, both theoretically and numerically. The discrete PINNs are thought to be time-consuming and seldom applied in the PINNs literature. In our paper, we have reduced the computational time with the transfer learning techniques and given an error estimate result. For comparison to related works like Runge-Kutta, classical numerical methods such as explicit Runge-Kutta suffer from stability constraints in time step and space step sizes, and implicit Runge-Kutta methods suffer from solving nonlinear equations many times, while our method is stable and easy to approximate in space by neural networks in every time stamp.
>
> **W2. Rationale and Numerical Experiments.** We insist that it is unfair to compare the solving time of PINNs and traditional numerical methods. Traditional methods have been carefully designed for decades while PINNs are still developing. PINNs and their variants also have advantages in problems with complex geometries and parametric PDEs. The main contribution of our paper is the progress made on the PINN communities.
>
> **Q1. Transfer learning clarification.** All the parameters are fine-tuned between two adjacent timestamps. For the simple RD Eq., solely updating the weights in the last hidden layer per time step is also enough to achieve high accuracy.
>
> **Q2. Can you choose large time-steps for stiff problems?** Our method shares the advantages of implicit time differencing schemes in the theory of traditional numerical methods, thus can choose large time-steps for stiff problems.

---

> > ### Comment · Reviewer_HW8p · 2023-11-19
> > **Reply to the Authors' Rebuttal**
> >
> > At the outset, I thank the authors for their brief reply. I am not at all convinced by any of the replies and I outline why below:
> >
> > 1. **Novelty**: I still don't see what is novel about replacing implicit RK2 as the ODE solver in the Raissi et. al. paper with Crank Nicolson -- no nonlinear equations are solved by Raissi et. al PINNs paper either as they also minimize a neural network loss with gradient descent at every time step. You are simply replacing one ODE solver by another.
> >
> > 2. **Comparison with Traditional Solvers**: Why should anyone use PINNs if they are orders of magnitude slower than classical numerical methods as I suspect is the case with all the benchmarks that you consider. I agree that PINNs have some advantages vis a vis complex geometries, high-dimensional problems, parametric PDEs etc. The issue is that you do not consider any of these problems in your paper.
> >
> > 3. **Choice of time step**: You do not answer the question that I asked you in this context. You should have performed the numerical experiments with the stiff problem that I suggested. Of course, in principle, you can take large time steps but then the solution also changes by quite a bit and your argument about transfer learning may not work.
> >
> > Given that the authors have not addressed any of my concerns, I retain my evaluation that the contributions of this article do not merit publication in this venue.

---

### Official Review · Reviewer_rdRW · 2023-10-31

**Soundness:** 3 good
**Presentation:** 3 good
**Contribution:** 2 fair
**Rating:** 5
**Confidence:** 5

**Summary:**

This paper deals with training of PINNs for transient PDEs. One major challenge in training PINNs for such evolutionary systems is that PINNs may violate temporal causality, i.e. the model may not be sufficiently trained at a given time step before approximating the next time steps. This is because the collocation points for PINNs are all trained simultaneously. The authors propose transfer learning enhanced discrete physics-informed neural networks (TL-DPINN) which has two key contributions: i) by using implicit time discretization the algorithm resolves temporal differentiation, ii) by using the transfer learning the algorithm accelerates the PINN training. In theorem 4.1, they show that, under certain assumptions, the error of training defined as the difference of ground truth solution (of dissipative PDEs) at a given time and that of the neural network is bounded. Four examples are considered, which progressively becomes more challenging, and several time-discretization schemes are compared.

**Strengths:**

The examples provided are involved enough to show the strength of the proposed algorithm. Example 1 and 2, although only 1D, are demonstrating the solution of nonlinear PDE. KS equation can become chaotic, so the success of TL-DPINN makes it a meaningful contribution to the community (although we still see some struggles after t=0.8). The NS example is in turbulent regime which poses a nontrivial problem for PINNs.
Section 5.5 is also a nice and careful addition of the paper. Comparison of SOTA plus considering several time scheming is a valid scientific work and is considered as a strength.

**Weaknesses:**

Although the theorem is a nice addition to the paper, the underlying assumptions seem too restrictive. In the domain of PDE analysis, I don't see it as a "weakness" (specially for nonlinear PDEs, its a daunting task, and often impossible, to come up with a generic theoretical framework) but more discussion is needed about the limitation of the work. What are the implications of assumption 1, and when do they fail (as an example)? Only a brief addition can be helpful (more details can be discussed in the appendix).

A major contribution of the paper is its success in the solution of chaotic PDEs. Some discussion in A.5.1 about limitations can be moved to the proper text to make this work more useful (and fair) for readers. From the results, seems like still we have room for improvement for such complicated scenarios, but this paper can have its own place; just need a further discussion about weaknesses and limitations (so the claims need to be revisited).

Also, the use of implicit time discretization is not novel. Authors mention it has been used for ODEs, so the application to PDEs is a fair contribution but is this the only novelty? Moreover, the use of transfer learning for speeding up the convergence is nice but there has been some works (for example using parallel in time algorithms for PINNs) to speed up the convergence for evolutionary PDEs. The combination of the two ideas seem like a novel algorithm but hopefully authors can clarify on the novelties further in the revised version.

Overall, I'm positive and would be happy, after addressing my concerns above, to update my score. Overall, I see this paper as a practical improvement for PINNs, rather than theoretical, but it has a good skeleton (because of nice examples and clear case studies).

**Questions:**

More like a suggestion: are authors familiar with NIF (https://arxiv.org/pdf/2204.03216.pdf). They treat time and space separately via shapeNet and paramterNet (they consider time like a parameter). Of course this is a different method, but gives the opportunity for treating time in a continuous way. I appreciate if authors open up the conversation on how they see discrete versus continuous treatment of time will impact the solution of evolutionary PDEs. Of course they can argue the original PINNs treat time as "continuous" but it comes with cost of causality issue but there might be solutions for this that are not discrete time version. It will be good for the community to start a conversation two approaches (from practical and theoretical point of views). Another similar work: https://arxiv.org/pdf/2206.02607.pdf

How would you comment on using transfer learning for original PINNs (without discrete time analysis)? No need for producing any results.

I couldn't find the discussion on how to pick the weights (\lambda values). How do authors tune such values? Do they use static values or are they updating such weights adaptively. It is important to add a discussion on how to pick such values.

Authors argue that Wang et all 2022a suffer from sensitivity to epsilon on page 2. One could argue that their approach this approach also introduces its own hyper-parameters such as Nt and M0, M1, etc. How do authors justify that?

To confirm, \matcal{L}^0(\theta^0) is simply learning the initial conditions?

 Please clarify the implications of assumptions 4.1 and 4.2. Also the text for theorem 4.1 needs some revising (e.g. there is no equation 4.1, etc.). The statement on page 5 "The second assumption can be verified by the standard regularity estimate result of PDEs Evans
(2022), and we omit it here for brevity." can be clarified in Appendix.

---

> ### Author Response · Authors · 2023-11-15
>
> **W1. The assumption 1 of the theorem.** Thanks for the positive comment and suggestion. This dissipative assumption is only used to prove the error bound Eq.(6). The error bound may also hold for non-dissipative equations. Our experimental results show that the method also works well for non-dissipative equations, such as the chaotic KS and NS eq.
>
> **W2. Chaotic PDEs discussion.** We believe that our method has the potential to successfully solve chaotic PDEs in a limited time region. The reason why it failed for large-time prediction may be that the accuracy of neural network approximation to the spatial PDE (Eq.(4)) in every timestamp is practically limited to $10^{-6}~10^{-8}$, while traditional spectral methods can have an accuracy of $10^{-12}~10^{-16}$. A small error in a previous time may cause a huge error at a later time due to the chaotic property.
>
> **W3. Novelties of the paper.** The novelty of the paper is that we contribute to the PINN community by rediscovering the good performance of the discrete PINNs applied to solve evolutionary PDEs, both theoretically and numerically. The discrete PINNs are thought to be time-consuming and seldom applied in the PINNs literature. In our paper, we have reduced the computational time with the transfer learning techniques and given an error estimate result. For comparison to related works like Runge-Kutta, classical numerical methods such as explicit Runge-Kutta suffer from stability constraints in time step and space step sizes, and implicit Runge-Kutta methods suffer from solving nonlinear equations many times, while our method is stable and easy to approximate in space by neural networks in every time stamp.
>
> **Q1. Discussion on discrete versus continuous treatment of time.** Thanks for the valuable suggestion about the concept of NIF. As the evolutionary PDEs are naturally treated from previous time to later time, the traditional numerical finite differencing scheme also possesses this property called time causality. We believe that the discrete-time version is more easily adapted to time causality while the continuous-time version is more convenient to use. I’m not sure if NIF treatment of continuous-time has the issue of causality or not at present. It is expected to have a conversation on the two approaches for the PINN community.
>
> **Q2. Comment on using transfer learning for original PINNs.** There’s a lot of work on using transfer learning for PINNs as we stated in the related work section. For original PINNs, TL is more likely to be used for the same class PDEs with different parameters or different initial/boundary conditions. This transfer-learning strategy is also applicable to our method.
>
> **Q3. How to pick the weights in the loss.** There are works to choose the weights adaptively to speed up the training process. However, since the loss form for other baselines is different, we simply pick the weights to be a constant (e.g. 1, 100) in all the training
>
> **Q4. Hyper-parameters sensitivity.** We acknowledge that our algorithm is also sensitive to the hyperparameters like causal PINNs. However, numerical results show that the results on optimal hyperparameters are better than causal PINNs, especially the training time cost.
>
> **Q5. \matcal{L}^0(\theta^0) is simply learning the initial conditions?** Yes. We’ll correct the claims, descriptions, and typo errors in the revised version.

---

### Official Review · Reviewer_sf3Z · 2023-11-08

**Soundness:** 3 good
**Presentation:** 3 good
**Contribution:** 3 good
**Rating:** 5
**Confidence:** 3

**Summary:**

The paper proposes a method to improve the training of Physics-informed neural networks (PINNs) for solving time-evolving partial differential equations (PDEs) that exhibit complex behaviors. By incorporating implicit time differencing and transfer learning, the method enforces temporal causality and efficiently updates the PINNs for different time frames. Specifically, instead of learning just one network for the whole time evolution, the paper proposes to first fix time steps, and then learn the update network at each time step. Empirical results show improvements on the accuracy and efficiency of PINNs with convergence proven under certain conditions.

**Strengths:**

The problem trying to solve is interesting, enforcing causality in PINNs is essentially providing more physical insights for the problem. The empirical improvement over the causal PINN in training time is significant, there are cases where the PINNs can not solve the PDE but this one can, and the theoretical results are sound.

**Weaknesses:**

The method achieves causality by fixing a time step and train separate networks for each time step. Despite the good performances, this constraints the PINNs to be meshless on time scale. Besides, training and saving many networks is expensive, especially for long time predictions.

I understand that transfer learning helps mitigate this problem, and Table 1 shows that the training time is lower than other methods, but it would be very helpful if the author can provide a figure, where the x-axis is the number of time steps or the max time for prediction, and the y-axis is the training time and logits, so we can have a better idea as to when does the proposed method becomes too cumbersome. I am willing to raise the score if this question is properly addressed.

Also, does it require more data to train the proposed method than vanilla PINNs as we are training much more parameters? If so, how much more?

**Questions:**

The main concern is on the scalability of the proposed method. Specifically, the method mentions that transfer learning works well when dt is small, can you specify how small is that, and does this change for different PDEs? Also, how does this method work for higher-dimensional PDEs?

---

> ### Author Response · Authors · 2023-11-15
>
> **W1. Training time comparison.** We give a comparison of the RD Eq here. In Table 1 the training time is $1463s$ for $200$ time steps. If we fix the time interval [0, 1] and increase the time steps to $400,600,800,1000$, the training time is $1984s,2339s,2743s,2977s$. The total training time increased slightly when the time steps increased. However, we must admit that the hyperparameters in Algorithm 1 for different time steps are also different. The limitation of our method is that the hyperparameter threshold value ϵ can affect the training time and accuracy greatly.
>
> **W2. Required data.** No, our method does not require additional data than vanilla PINNs. We only need the initial/boundary data as the data needed for vanilla PINNs.
>
> **Q1. Scalability of dt.** We insist that the $\Delta t$ should be small to get a small prediction error, as given in our paper's error bound Eq.(6). Large $\Delta t$ will lead to large prediction errors in time, and the training will be nonsense. The value of $\Delta t$ is chosen similar to that in traditional finite difference methods.

---

> > ### Comment · Reviewer_sf3Z · 2023-11-20
> >
> > Thank you for the response. I will keep my score as is.

---

### Official Review · Reviewer_XCvN · 2023-11-09

**Soundness:** 1 poor
**Presentation:** 2 fair
**Contribution:** 1 poor
**Rating:** 3
**Confidence:** 5

**Summary:**

In this paper authors propose a discrete PINNs model for solving evolutionary partial differential equations with multi-scale or turbulent behavior over time. The method uses implicit time differencing schemes to enforce temporal causality. In addition, the method inherites ideas from transfer learning methods to sequentially update the PINNs in space as surrogates for PDE solutions in different time frames. A variety of benchmarking problems, as well as ablation studies were considered to evaluate the performance of the proposed method, and for some test cases the results were compared with a few PINNs baseline methods.

**Strengths:**

- The attempt of making PINNs model more robust for efficient and accurate prediction of evolutionary/stochastic behaviors sounds interesting.
- extensive numerical experimentations is also appreciated.
- The code and the input data are shared which is helpful for benchmarking and reproducing the presented results.

**Weaknesses:**

There are some key missing factors which make me not recommend the publication of this paper. Below I mention just the key points and I skip the ones with less priority for me:
- the transfer learning proposal is not new idea for SciML problems in general, and the method utilized in this paper (fine tuning only last layers) has been used in other papers as well. Authors have cited a few related works, there are more examples which are closely related, checkout the following papers for example :
  - Towards Foundation Models for Scientific Machine Learning:Characterizing Scaling and Transfer Behavior (https://arxiv.org/pdf/2306.00258.pdf)
  - The Old and the New: Can Physics-Informed Deep-Learning Replace Traditional Linear Solvers? (https://arxiv.org/abs/2103.09655)
  - Transfer Learning Enhanced DeepONet for Long-Time Prediction of Evolution Equations (https://www.semanticscholar.org/paper/Transfer-Learning-Enhanced-DeepONet-for-Long-Time-Xu-Lu/a2ab63a272fa0762a75eb3d1b0a86506a9db3623)

- Following the previous point, its effectiveness of the transfer learning in this work heavily relies on the choice of $\Delta t$. Authors qualitatively mentioned that "if a small enough time step is used, there are little difference between two adjacent network parameters, $\theta^n$ and $\theta^{n+1}$". However, as far as I realized there is no concrete discussions/numerical experiments provided to show how much the time-step parameter could be increased and where the applicability of fine-tuning last layers could break. Not addressing such points and addressing things qualitatively and at the surface level questions the quality and the comprehensiveness of the research conducted.

- In theoretical result, authors made the assumption that the equation must be dissipative, which I did not quite get this part. First of all what authors mean by equations being dissipative? does it mean the equations always require dissipative/diffusive terms? It also makes me wonder whether this method is applicable for a general hyperbolic equations potentially with some external forces, such as high Reynolds regime flows?

- The presentation of the paper needs significant improvement. For example, some of result visualizations and discussion are not informative, for example what authors try to convey by presenting Figure(3)? in particular for (b) how does it show that their method reduces the number of epochs required at each time-step, because overall it requires more than 500 epochs and for intermediate steps it jumps to much larger numbers? There are multiple examples like this in the main body of the paper where the results were not discussed clearly.

- I did not see that authors discuss any limitations or challenges associated with the proposed method. Note that there is no free lunch :) A good quality science work needs to discuss different aspects of the proposed method including the drawback. I encourage authors to share these types of contents in their science publications.

**Questions:**

- Authors mentioned in Appendix that a multi-head neural network may lead to slow convergence, which is not clear to me why? Having a multi-head single neural network could be beneficial in terms of sharing parameter and reduce the computational effort.
- For the MLP network considered in this work, I wonder how authors decide to use $tanh(.)$ for activation functions? it would be useful to provide ablation study to numerically provide insight why $tanh(.)$ was selected.
- Regarding Fourier Features, which were justified for periodic and oscillatory behaviors, I wonder what about problems that do not involve such behavior and whether these types of feature embedding could negatively impact the results? It would be great if authors could provide a clear discussion on this.
- Since authors literally do time integration numerically, I wonder if they numerically test their numerical accuracy of their prediction and whether it is second order for Crank-Nicolson scheme?
- The experimental setting is not quite clear. For their evaluations did authors consider out of domain data (data outside training range), or the focus was on in-domain evaluation? Also, did they explore whether their method could help for long term predictions (such as weather forecasting problems?)?

**Details Of Ethics Concerns:**

No concern.

---

> ### Author Response · Authors · 2023-11-15
>
> **W1. Transfer learning works.** Thanks for offering TL-related works in SciML problems. There’re many related works so we only cited some PINN-related works in our paper. We’ll cite more in the revised version. Moreover, we have cited and discussed the work “Transfer Learning Enhanced DeepONet…” in Line 10-14 Page 3 of our paper.
>
> **W2. How much the time-step parameter could be increased to break the fine-tuning of the last layer?** We insist that the $\Delta t$ should be small to get a small prediction error, as given in our paper's error bound Eq.(6). Large $\Delta t$ will lead to large prediction errors in time, and the training will be nonsense. The value of $\Delta t$ is chosen similar to that in traditional finite difference methods. In Appendix A.4.2 we show that the total parameters and computation of all discrete PINNs are less or about the same with a single continuous PINN.
>
> **W3. Dissipative assumption.** This assumption is only used to prove the error bound Eq.(6). Experimental results show that the method also works well for non-dissipative equations, such as the chaotic KS and NS eq.
>
> **W4. Explanation like Figure(3)(b).** Figure(3)(b) is used to state that each timestamp’s training epoch is small across the time domain, reducing the total training time. We revised the statement to make it readable.
>
> **W5. Limitations or challenges.** Thanks for your kind suggestion. The limitation is that the hyperparameters affect the training accuracy and time. We’ll implement the section before the conclusion.
>
> **Q1. Multi-head neural networks.** Theoretically, a single multi-head neural network to represent the $q$ hidden stage and $u^{n+1}$ could be beneficial in terms of sharing parameters and reducing the computational effort. However, we experimentally found that it needs more epochs and more time to train to the same loss compared to using $q+1$ neural networks separately.
>
> **Q2. Using tanh for activation functions.** The most commonly used relu activation function isn’t suitable for PINN approximation since the second-order derivative will be zero and doesn’t contribute to the high-order term in residual loss of the partial differential equations. Nonlinear tanh is a common choice in the PINN community.
>
> **Q3. Fourier Features.** Thanks for the comment. Adding Fourier features in the embedding layer can speed convergence for smooth periodic or oscillatory functions. However, like the numerical spectral methods, it may lead to slow convergence for nonsmooth or discontinuous functions.
>
> **Q4. Numerical integration in time.** We theoretically proved that it is second-order in time for the Crank-Nicolson scheme in the error bound Eq.(6), i.e. the term $\tau^2$. If we train the loss in every timestamp small and the collation points large enough, the lead term of the error bound will be $\tau^2$. We numerically verify it by computing RD Eq. for $\Delta t=1/100$ and $1/200$, the error is $7.01e-05$ and $1.82e-05$ respectively.
>
> **Q5. Experimental settings.** We clarify that our discrete PINNs can predict at trained discrete timestamps in time and arbitrary continuous physical points in space. For long-time prediction, we can train it iteratively in time to predict.